# Risk and protective factors for self-harm and suicide behaviours among serving and ex-serving personnel of the UK Armed Forces, Canadian Armed Forces, Australian Defence Force and New Zealand Defence Force: A systematic review

Charlotte Williamson[1]*, Bethany Croak[1], Amos Simms[2,3], Nicola T. Fear[1,2], Marie-Louise Sharp[1‡], Sharon A. M. Stevelink[1,4‡]

1 King's Centre for Military Health Research, King's College London, London, United Kingdom, 2 Academic Department of Military Mental Health, King's College London, London, United Kingdom, 3 British Army, London, United Kingdom, 4 Department of Psychological Medicine, King's College London, London, United Kingdom

‡ These authors contributed equally as joint last authors.

* charlotte.1.williamson@kcl.ac.uk

## Abstract

### Background

Self-harm and suicide behaviours are a major public health concern. Several factors are associated with these behaviours among military communities. Identifying these factors may have important implications for policy and clinical services. The aim of this review was to identify the risk and protective factors associated with self-harm and suicide behaviours among serving and ex-serving personnel of the United Kingdom Armed Forces, Canadian Armed Forces, Australian Defence Force and New Zealand Defence Force.

### Methods

A systematic search of seven online databases (PubMed, Web of Science, Embase, Global Health, PsycINFO, PTSDpubs and CINAHL) was conducted alongside cross-referencing, in October 2022. Following an *a priori* PROSPERO approved protocol (CRD42022348867), papers were independently screened and assessed for quality. Data were synthesised using a narrative approach.

### Results

Overall, 28 papers were included: 13 from Canada, 10 from the United Kingdom, five from Australia and none from New Zealand. Identified risk factors included being single/ex-relationship, early service leavers, shorter length of service (but not necessarily early service leavers), junior ranks, exposure to deployment-related traumatic events, physical and

**Data Availability Statement:** Data availability is not applicable to this article as no new data were created or analysed in this study.

**Funding:** This work is part of a PhD nested within Phase 4 of the King's Centre for Military Health Research Health and Wellbeing Cohort Study and funded by the Office of Veterans' Affairs, Cabinet Office, UK Government [Contract ref: CCZZ20A88] (https://www.gov.uk/government/organisations/office-for-veterans-affairs). NTF and SAMS secured the funding for Phase 4 of the King's Centre for Military Health Research Health and Wellbeing Cohort Study, and CW is currently in receipt of the funded PhD studentship. The funder played no role in the study design, data collection and analysis, decision to publish, or preparation of the manuscript.

**Competing interests:** CW is currently in receipt of a funded PhD studentship via Phase 4 of the King's Centre for Military Health Research Health and Wellbeing Cohort Study funded by the Office for Veterans' Affairs (OVA), Cabinet Office, UK Government. BC is currently in receipt of a funded PhD studentship from The Economic and Social Research Council (ESRC), UK Government. AS is a full-time member of the British Army seconded to King's College London. NTF is partly funded by the United Kingdom's Ministry of Defence (MOD). MLS is fully funded by a grant from the OVA. SAMS is supported by the National Institute for Health and Care Research (NIHR) Maudsley Biomedical Research Centre at South London and Maudsley NHS Foundation Trust and the National Institute for Health and Care Research, NIHR Advanced Fellowship, Dr Sharon Stevelink, NIHR300592. The views expressed in this publication are those of the authors and not necessarily those of the OVA, the ESRC, the British Army, the MOD, the NHS, or the NIHR. All authors have approved the manuscript and agree with its submission to PLOS ONE. Any conflicts of interest have been outlined in the manuscript. This does not alter our adherence to PLOS ONE policies on sharing data and materials.

mental health diagnoses, and experience of childhood adversity. Protective factors included being married/in a relationship, higher educational attainment, employment, senior ranks, and higher levels of perceived social support.

## Conclusion

Adequate care and support are a necessity for the military community. Prevention and intervention strategies for self-harm and suicide behaviours may be introduced early and may promote social networks as a key source of support. This review found a paucity of peer-reviewed research within some populations. More peer-reviewed research is needed, particularly among these populations where current work is limited, and regarding modifiable risk and protective factors.

## 1. Introduction

Self-harm and suicide behaviours are a major public health concern, with over 700,000 people dying by suicide and an estimated 14.6 million people engaging in self-harm each year globally [1,2]. The aetiology and onset of these behaviours is complex and multifaceted; prevalence and risk are influenced by several factors including age, sex, ethnicity, geographical region and occupation [3–5]. Military populations are a potentially at-risk group as military service impacts on the health and wellbeing of personnel during and after service [6]. Self-harm and death by suicide appear to be relatively rare among the military community, although rates of both behaviours have increased in recent years but remain either lower or comparable to the general population [7–12]. Suicide behaviours can present as ideation (i.e. thoughts) and attempts which are also prevalent among serving and ex-serving personnel; a reported 11% global prevalence of these behaviours in the entire military [13], with prevalence varying slightly between serving and ex-serving personnel for both suicidal ideation (serving = 10%; ex-serving = 14%) and suicide attempts (serving = 8%; ex-serving = 15%) [13].

Previous research has reported that serving and ex-serving personnel with mental health diagnoses are at risk of self-harm and suicide behaviours, for example post-traumatic stress disorder (PTSD) [14–18]. Another specific at-risk group includes those separated from the military [19–21], especially if separation occurred involuntarily (e.g. medical discharge) [22]. Additionally, younger age is a risk factor [22–24]; a recent UK study reported suicide risk was two-to-three times higher in men and women under 25 years old who had left service (vs the general population) [24]. Ex-serving personnel with a shorter length of service (<10 years of service) were also at increased risk [24]. Protective factors have been less frequently explored, but include social support (i.e. the availability and adequacy of social connections [25]) [26,27], higher educational attainment [28], employment [29] and holding a more senior rank (e.g. officer) [15,18].

There remains a paucity of research exploring the risk and protective factors for self-harm and suicide behaviours among several military populations, with available literature focussing on a limited number of countries. This review focused on military populations from four nations of the Five Eyes intelligence alliance [30]: namely the UK, Canada, Australia and New Zealand. The Five Eyes nations (which also includes the United States (US)) have a combined military population estimated at 2.6 million, are all developed countries which share a common language, have similarities in society and culture, broadly similar military involvement,

and well-resourced military healthcare systems [31]. Of the limited systematic reviews in the field, studies on US military populations make up all or the majority of included papers [27,32,33]. To address this gap in the literature, the current review focuses on the remaining four Five Eyes nations where attention has been limited previously. Additionally, the US was excluded due to differences in firearms culture, including access, licensing and laws which affects access to means of suicide; a recent systematic review highlighted that US veterans are at substantially increased risk of firearm suicide and have higher rates of firearm ownership than the US general population [34].

The current systematic review aimed to identify risk and protective factors associated with self-harm and suicide behaviours among serving and ex-serving personnel of the UK Armed Forces, Canadian Armed Forces, Australian Defence Force and New Zealand Defence Force.

## 2. Method

### 2.1. Design

This systematic review was conducted following Preferred Reporting Items for Systematic Reviews and Meta-Analyses (PRISMA) guidelines [35]. Prior to commencing the review, the protocol was registered with PROSPERO (CRD42022348867), an online database of systematic review protocols submitted prospectively to maintain research integrity.

### 2.2. Search strategy

Seven electronic databases were searched in October 2022: PubMed (including MEDLINE and PubMed Central), Web of Science, Embase, Global Health, PsycINFO, PTSDpubs and CINAHL. All databases were searched using pre-defined terms related to: (1) self-harm and suicide behaviours, (2) the military, and (3) geographical locations. See **S1 Table in S1 File** for full search strategy.

The search included all original, peer-reviewed work that identified risk and/or protective factors associated with self-harm and/or suicide behaviours both during (serving personnel) and after (ex-serving personnel/veterans) military service. Restrictions were placed on publication dates from 1st January 2001 to 30th September 2022 to allow for the coverage of the start of the Iraq and Afghanistan conflicts. Additionally, these limits were chosen due to better availability and quality of literature in the field of military medicine since 2001 [36].

PROSPERO was searched to identify any ongoing systematic reviews and meta-analyses of relevance. Additionally, the reference lists of included studies and other relevant studies were searched, including identified systematic reviews and meta-analyses of relevance. If the full-text of a paper was not available online, authors were contacted for access. At least one expert in the field of military self-harm and/or suicide behaviour research from each nation was contacted to ensure key papers had been identified. Experts were identified via the Five Eyes Mental Health Research and Innovation Collaboration [37].

### 2.3. Study selection criteria

Eligibility was determined using the following criteria:

- Published in English

- Original, peer-reviewed work

- Published 1st January 2001 to 30th September 2022

- Studies that reported risk and/or protective factors for outcomes of self-harm, suicidal ideation, suicide attempts and/or completed suicide

- Population comprising of serving and/or ex-serving personnel (Regulars/Reservists; Navy/Army/Air Force/Marines) from the UK Armed Forces, Canadian Armed Forces, Australian Defence Force or New Zealand Defence Force

- If military only made up part of the reported sample, only papers that reported on outcomes for serving/ex-serving personnel separate from the other population (e.g., general population) were included

It is important to note the term 'veteran' is defined and used differently across nations. For instance, in the UK a veteran is defined as someone who has served a minimum of one day paid employment in the UK Armed Forces but no longer serves [38], this can include those who deployed or not and is irrespective of their type of discharge. However, in Canada, a veteran is any former member of the Canadian Armed Forces who successfully underwent basic training and is honourably discharged [39]. Therefore, for the purpose of this review, we adopted the term 'ex-serving personnel' to mean someone who served in the Armed Forces but no longer serves, irrespective of deployment experience or type of discharge.

Exclusion criteria included:

- Study design: qualitative, systematic reviews/meta-analyses, pilot studies, case-control studies, case reports/series, study protocols, grey literature, conference abstracts/papers, dissertations/theses, and editorials

- Outcomes of assisted suicide (i.e. the act of deliberately assisting another person to kill themselves [40]) and suicide bombing (i.e. a terrorist bomb attack in which the perpetrator expects to die whilst killing a number of other people)

- Studies that only included a population sample of conscripts (i.e., people compulsorily enlisted into the military), cadets or officer students

## 2.4. Screening and data extraction

Following the initial search, all identified studies were imported into Endnote 20 and duplicates were removed. CW independently reviewed the titles/abstracts of all papers. Subsequently, the full papers for the remaining studies identified as relevant were then reviewed. The reference lists of all included papers were manually searched for any additional papers of relevance (i.e., cross-referencing). BC independently performed second reviewer screening on a sample of studies (10% at title/abstract screening stage and 20% at full text screening stage). The reviewers (CW and BC) independently decided which studies met the eligibility criteria to be included in the review and, at full text screening stage, noted any reasons for exclusion. Any discrepancies were resolved through discussion. Interrater reliability was calculated at each screening stage; agreement was 99% at title/abstract and 100% at full-text stage.

The following data were extracted independently by CW for all included papers where available:

- **General information:** title, lead author, publication date, journal title, location/country of study

- **Study characteristics:** study aim, study design and methods, response rate *(where relevant)*, sample size, data collection date

- **Sample characteristics**: age in years *(mean, median or range)*, sex distribution, ethnicity, population type (e.g., clinical/non-clinical)

- **Military characteristics:** serving status (serving/ex-serving), engagement type (regular/reserve), service branch (Army, Naval Services, Air Force), rank (other, non-commissioned, commissioned), deployment experience (e.g., number and duration of deployments), era of service (e.g., Iraq and Afghanistan era)

- **Outcomes:** outcome type (i.e., self-harm, suicidal ideation, suicide attempts, completed suicide), definition of outcome, how outcome was measured

- **Risk and protective factors** associated with self-harm and/or suicide behaviour outcomes

- **Study findings:** conclusions, limitations, future research

Statistical findings are reported in **S2 Table in S1 File**. Where available, adjusted odds ratios or effect estimates have been presented. Otherwise unadjusted results have been reported and clearly identified.

## 2.5. Data synthesis

To examine the risk and protective factors associated with self-harm and suicide behaviours, a narrative synthesis was performed. This approach was chosen due to heterogeneity across studies, for example, variation in sample sizes, target populations and outcomes [41]. Due to variation across included studies, such as differences in study design and outcome measurements, and the large number of risk and protective factors identified, it was neither practical nor feasible to conduct a meta-analysis as part of this review.

## 2.6. Quality assessment

The quality of each included paper was assessed independently by CW using the National Heart, Blood and Lung Institute (NHBLI) tailored quality assessment tools [42]. BC independently performed second reviewer quality assessment on a sample of papers (20%). Any discrepancies were resolved through discussion. Papers were not excluded based on their quality, instead, results from the assessment provided additional insight into the quality of research in the field.

# 3. Results

Overall, 4,576 papers were identified, of which 497 duplicates were removed (**Fig 1**). The title/abstracts of 4,079 papers were screened, leaving 94 papers at full-text stage. Experts in the field identified no additional peer-reviewed papers and cross-referencing identified one additional paper of relevance. Overall, 28 papers met the inclusion criteria (**S3 Table in S1 File**).

## 3.1.Study characteristics

Included studies used a range of study designs; 23 cohort studies [15,18,28,29,43–61] and five retrospective cohort studies [62–66]. The majority of papers were from Canada (n = 13) [43,45–47,50–52,56–58,65,66] and the UK (n = 10) [15,18,53,54,59–64]. The remaining papers were from Australia (n = 5) [28,29,48,49,55], but none were from New Zealand. Papers explored military samples of ex-serving personnel only (n = 12) [45–48,50,56–58], serving personnel only (n = 8) [28,29,49,51–54,59,62–64,66] and mixed samples of serving and ex-serving personnel (n = 8) [15,18,43,44,55,60,65,67]. Of the 28 included papers, 24 included mixed male and female samples, however the percentage of male participants typically ranged from around 85% to 99% [15,18,29,43,45–47,50–66]. Of the remaining four papers, three used all

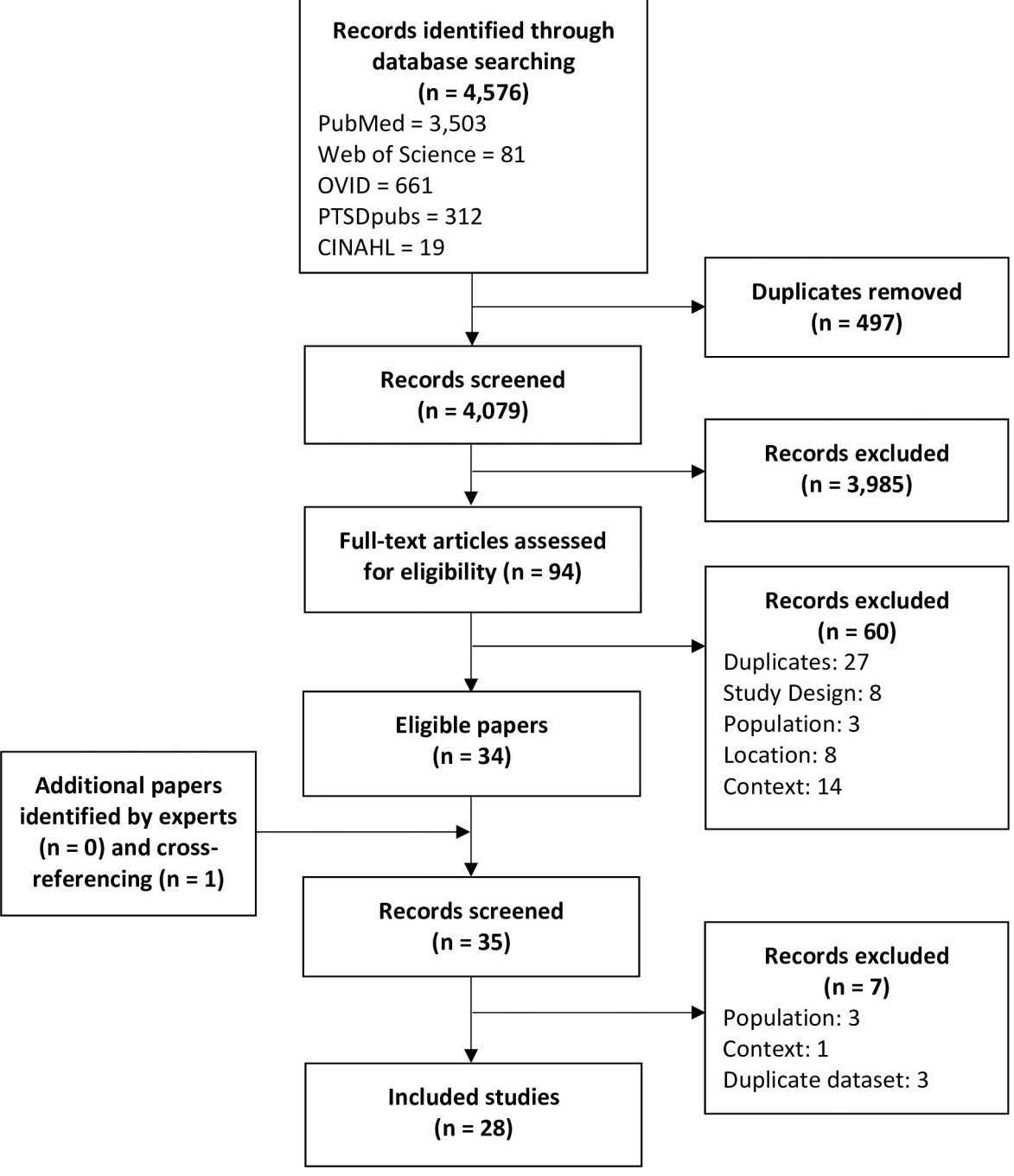

**Fig 1. PRISMA flow diagram.**

male samples [28,48,49] and one did not explicitly report on the sex of the sample directly within the paper [44].

### 3.2. Quality assessment

Most papers received a quality assessment score of 'fair' (n = 21), and the rest scored 'good' (n = 7) (**S4 Table in S1 File**). Common reasons for lower scores included lacking a sample size

justification/power description, not assessing the exposure variables more than once, and not using outcome measures which were clearly defined, valid and reliable.

### 3.3. Reported self-harm and suicide behaviour outcomes

Included papers reported on a range of outcomes outlined in Table 1. Four papers reported on self-harm only [15,60–62], nine reported on suicidal ideation only [43–45,51–53,58,59,65], two on suicide attempts only [29,57] and three on completed suicides only [63,64,66]. The remaining 10 papers reported on a mix of these outcomes, typically referred to as 'suicidality' (often including outcomes of suicidal ideation, suicide plans and suicide attempts) [18,28,46–50,54–56].

Table 1. Characteristics of included studies.

| STUDY | | SAMPLE | | | | OUTCOME | | |
|---|---|---|---|---|---|---|---|---|
| Study and Location | Study Design | Military Sample Size | Demographic and Military Characteristics | Male (%) | Age | Outcome Type | Outcome Definitions | Outcome Measures |
| Afifi et al (2016)[+1] [56] *Canada* | Cohort study | 8,161 | Serving Canadian Armed Forces personnel—Regular (n = 6,692) and Reserve (n = 1,469) Various military ranks including junior non-commissioned member (NCM), senior NCM and officer Deployed (46% of Regulars and 100% of Reservists) and non-deployed (54% of Regulars and no Reservists) | **Regulars:** 86.1% **Reserves:** 90.6% | 18 to 60 years | Suicidal ideation, suicide plans and suicide attempts | Not reported | **Suicidal Ideation:** Ever thought seriously about committing suicide or taking his or her own life (yes/no); in the past year (yes/no) **Suicide Plan:** Ever made a plan to commit suicide (yes/no); in the past year (yes/no) **Suicide Attempt:** Ever attempted suicide or tried to take own life (yes/no) |
| Belik et al (2009)[+2] [57] *Canada* | Cohort study | 8,441 **Regular:** 5,155 **Reserves:** 3,286 | Serving Canadian Armed Forces personnel—Regular (n = 5,155) and Reserve (n = 3,286) Regions of Atlantic, Quebec, Ontario and Prairies Various military ranks including junior NCM, senior NCM and officer | Male and female included but % not reported | 16 to 54 years | Suicide attempts | Not reported | Ever attempted suicide or tried to take own life (yes/no) |
| Bergman et al (2019)[+3] [62] *UK* | Retrospective cohort study | 56,205 | Ex-serving UK (Scottish) Armed Forces personnel (Regular only) Born between 1945 and 1985, resident in Scotland and registered with NHS Scotland both before and after service | 90.7% | *Mean age of first self-harm episode:* **Male:** 41.9 years **Female:** 38.9 years | Self-harm | First outcome of ICD-10 code X60-X84 *(intentional self-harm)*, Y87.0 *(sequelae of self-harm)*, Y10-Y34 *(event of undetermined intent)* or Y87.2 *(sequelae of events of undetermined intent)* or ICD-9 code E950-E959 *(suicide and self-inflicted injury)* or E980-E989 *(injury undetermined whether accidentally or purposely inflicted)* at any position in the record other than the cause of death, excluding any individual whose death was classified as suicide or whose date of death was the same as the recorded date of self-harm, whilst suicide was classified as the occurrence of any of the specified ICD codes in the cause of death field | ICD-10 code X60-X84 *(intentional self-harm)*, Y87.0 *(sequelae of self-harm)*, Y10-Y34 *(event of undetermined intent)* or Y87.2 *(sequelae of events of undetermined intent)* or ICD-9 code E950-E959 *(suicide and self-inflicted injury)* or E980-E989 *(injury undetermined whether accidentally or purposely inflicted)* [first outcome] |

*(Continued)*

**Table 1.** (Continued)

| STUDY | | SAMPLE | | | | OUTCOME | | |
|---|---|---|---|---|---|---|---|---|
| Study and Location | Study Design | Military Sample Size | Demographic and Military Characteristics | Male (%) | Age | Outcome Type | Outcome Definitions | Outcome Measures |
| **Bergman et al (2022)**[+3] [63] *UK* | Retrospective cohort study | 78,157 | Ex-serving UK (Scottish) Armed Forces personnel (Regular only) Born between 1945 and 1995, resident in Scotland and registered with NHS Scotland both before and after service | 90.3% | 18 to 80 years | Completed suicide | A cause of death recorded as ICD-10 codes X60–X84 *(intentional self-harm)*, Y87.0 *(sequelae of self-harm)*, Y10-Y34 *(event of undetermined intent)* or Y87.2 *(sequelae of events of undetermined intent)* or ICD-9 codes E950–E959 *(suicide and self-inflicted injury)* or E980–E989 *(injury undetermined whether accidentally or purposely inflicted)* [any position on death certificate]. Encompasses death resulting from suicide, intentional self-harm or events of undetermined intent | ICD-10 codes X60–X84 *(intentional self-harm)*, Y87.0 *(sequelae of self-harm)*, Y10-Y34 *(event of undetermined intent)* or Y87.2 *(sequelae of events of undetermined intent)* or ICD-9 codes E950–E959 *(suicide and self-inflicted injury)* or E980–E989 *(injury undetermined whether accidentally or purposely inflicted)* [any position on death certificate] |
| **Boulos & Fikretoglu (2018)**[+1] [58] *Canada* | Cohort study | 4,854 **Regular:** 3,385 **Reserves:** 1,469 | Serving Canadian personnel–Regular (n = 3385) and Reserve (n = 1469) All had deployed to Afghanistan Various military ranks including junior NCM, senior NCM and officer | **Regular:** 89.0% **Reserves:** 91.0% | 17 to 60 years | Suicidal ideation | Not reported | Have you thought about committing suicide in the past 12 months (yes/no) |
| **Harden & Murphy (2018)** [59] *UK* | Cohort study | 144 | Ex-serving UK Armed Forces personnel All were actively receiving treatment for mental health difficulties | 97.2% | <35 years = 18.8% 35–44 years = 27.8% 45–54 years = 27.4% ≥55 years = 27.1% | Suicidal ideation | Not reported | Clinical Risk assessments by psychiatric nurses: **Green:** no concerns raised by clinical staff, no plans, stable in terms of mental health presentation, may have history of suicidal ideation or low mood but no current thoughts and manages with routine support **Amber:** experiencing suicidal ideation but no plans were in place, suicidal ideation present but without consistent thoughts and evidence of protective factors **Red:** evidence of recurring suicidal ideation, evidence of plan and talks about committing suicide |
| **Hines et al (2013)**[+4] [60] *UK* | Cohort study | 9,803 | Serving (n = 7,567) and ex-serving (n = 2,213) UK Armed Forces personnel–Regular (n = 8,127) and Reserves (n = 1,676) Iraq and Afghanistan era Deployed (n = 6,598) and non-deployed (n = 3,205) | 88.1% | *Mean age:* **Self-harm:** 31.0 years | Self-harm | Not reported | Have you ever purposely harmed yourself (e.g. overdose)? (yes/no) |

(*Continued*)

**Table 1.** (*Continued*)

| STUDY | | SAMPLE | | | | OUTCOME | | |
|---|---|---|---|---|---|---|---|---|
| Study and Location | Study Design | Military Sample Size | Demographic and Military Characteristics | Male (%) | Age | Outcome Type | Outcome Definitions | Outcome Measures |
| Jones et al (2019)[+4] [18] *UK* | Cohort study | **Phase 1:** 10,272 **Phase 2:** 9,990 **Phase 3:** 8,581 **Interview Study:** 1,448 | Serving and ex-serving UK Armed Forces personnel (Regular and Reserves) **Phase 1:** Serving (n = 9,109) and ex-serving (n = 1,140) **Phase 2:** Serving (n = 7,141) and ex-serving (n = 1,804) **Phase 3:** Serving (n = 4,454) and ex-serving (n = 3,319) **Interview Study:** Serving (n = 809) and ex-serving (n = 639) Iraq and Afghanistan era Deployed and non-deployed | 85% | <30 years = 18.8% 30–34 years = 14.9% 35–39 years = 18.6% 40–49 years = 35.1% >49 years = 16.6% | Self-harm, suicidal ideation and suicide attempts | Not reported | **Self-Report Survey (Phases 1–3)** Have you ever purposely harmed yourself (e.g. overdose)? (yes/no) **Interview Study** **Suicide Attempt:** Ever made an attempt to take your life by taking an overdose of tablets or in some other way? (yes/no) **Suicidal Ideation:** Ever thought of taking your own life, even though you might not actually do it? (yes/no) **Self-Harm:** Ever deliberately harmed yourself in any way but not with the intention of killing yourself? (yes/no) **[Clinical Interview Schedule–Revised (CIS-R)]** |
| Jones et al (2021)[+4] [61] *UK* | Cohort study | 4,447 | Serving and ex-serving UK Armed Forces personnel (Regular only) **Junior entrants:** Serving (n = 644) and ex-serving (n = 553) **Standard entrants:** Serving (n = 2,171) and ex-serving (n = 1,079) Iraq and Afghanistan era Deployed and non-deployed **Junior entrants:** Deployed to Iraq/Afghanistan (n = 828); Deployed in combat role to Iraq/Afghanistan (n = 298) **Standard entrants:** Deployed to Iraq/Afghanistan (n = 2,238); Deployed in combat role to Iraq/ Afghanistan (n = 699) Junior entrants (joined services under 17.5 years; n = 1,197) and standard entrants (joined services at 17.5 years or older; n = 3,250) | *Junior entrants* 94.7% | *Median age:* **Junior entrants:** 35 years | Self-harm | Not reported | Have you ever purposely harmed yourself (e.g. overdose)? (yes/no) |
| Kapur et al (2009) [64] *UK* | Retrospective cohort study | 233,803 | Ex-serving UK Armed Forces personnel (Regular only) Attended the Military Service Trauma Recovery Day Program | 96% | *Median age of completed suicide:* 22 years | Completed suicide | Verdict of suicide (ICD-10 codes X60-X84 *[intentional self-harm]* and Y87.0 *[sequelae of intentional self-harm]*) or undetermined death/ open verdict (ICD-10 codes Y10-Y34 *[event of undetermined intent]*, excluding Y33.9 *[verdict pending]* and Y87.2 *[sequelae of events of undetermined intent]*) at inquest | Verdict of suicide (ICD-10 codes X60-X84 *[intentional self-harm]* and Y87.0 *[sequelae of intentional self-harm]*) or undetermined death/open verdict (ICD-10 codes Y10-Y34 *[event of undetermined intent]*, excluding Y33.9 *[verdict pending]* and Y87.2 *[sequelae of events of undetermined intent]*) at inquest |

(*Continued*)

**Table 1.** (Continued)

| STUDY | | SAMPLE | | | | OUTCOME | | |
|---|---|---|---|---|---|---|---|---|
| Study and Location | Study Design | Military Sample Size | Demographic and Military Characteristics | Male (%) | Age | Outcome Type | Outcome Definitions | Outcome Measures |
| **Kerr et al (2018)** [29] *Australia* | Cohort study | 229 | Ex-serving Australian Defence Force personnel (Regular only) Deployed and non-deployed Attending PTSD Recovery Program 2007–2014 | 99.1% | 24 to 86 years | Suicide attempts | Not reported | Have you made a previous attempt on your own life? (yes/no) |
| **Ketcheson et al (2018)** [43] *Canada* | Cohort study | 666 | Serving and ex-serving Canadian Armed Forces personnel Deployed and non-deployed Treatment-seeking at Operational Stress Injury Clinic | 90.2% | 18 to 64 years | Suicidal ideation | Not reported | In the last 2 weeks have you been bothered by thoughts that you would be better off dead, or hurting yourself in some way? (not at all to nearly every day) [**Patient Health Questionnaire (PHQ-9)**] |
| **Nelson et al (2011)**[+2] [44] *Canada* | Cohort study | 8,441 | Serving Canadian Armed Forces personnel–Regular (n = 5,155) and Reserves (n = 3,286) | Not reported | 16 to 84 years | Suicidal ideation | Not reported | In the past 12 months, did you seriously think about committing suicide or taking your own life? [**World Mental Health–Composite International Diagnostic Interview (WMH-CIDI)**] |
| **O'Toole et al (2015)** [28] *Australia* | Cohort study | 448 | Ex-serving Australian Defence Force personnel Ageing Australian Vietnam ex-serving Army personnel | 100% | *Mean age:* 60.4 years | Lifetime suicidality | Suicidality is any self-initiated behaviour occurring on a continuum ranging from suicidal ideations, to making a suicide plan, through to a suicide attempt [O'Carroll et al, 1996] | **Depression screener questions** *History of two weeks or more feeling sad, empty or depressed *Two weeks of loss interest in most things like work, hobbies, things usually enjoyed *If yes*: *Thoughts of death and thoughts of suicide (yes/no) *If yes, questions on suicide plans and suicide attempts |
| **Pinder et al (2012)**[+4] [15] *UK* | Cohort study | 821 | Serving and ex-serving UK Armed Forces personnel (Regular and Reserves) Deployed and non-deployed–Iraq **Intentional self-harm** Serving (n = 58) and ex-serving (n = 42); Deployed to Iraq (n = 65) and not deployed to Iraq (n = 35) **No intentional self-harm:** Serving (n = 545) and ex-serving (n = 176); Deployed to Iraq (n = 498) and not deployed to Iraq (n = 223) | *Intentional self-harm:* 5.6% *No intentional self-harm:* 94.4% | *Mean age;* **Intentional self-harm:** 34.40 years | Self-harm [Suicide attempts only prevalence] | Self-harm is 'self-poisoning or self-injury irrespective of the apparent purpose of the act' [National Clinical Practice Guideline] | **Self-Harm:** Ever deliberately harmed yourself but not with the intention of killing yourself? (yes/no) **Suicide Attempt:** Ever made an attempt to take your own life, by taking an overdose of tablets or in some other way (yes/no) [**Adult Psychiatric Morbidity Survey (APMS) 2000**] |
| **Richardson et al (2017)**[1] [45] *Canada* | Cohort study | 6,700 | Serving Canadian Armed Forces personnel—Regular only Deployed to Afghanistan at least once (45.1%) and non-deployed | 86.1% | *Mean age:* 35 years | Suicidal ideation | Not reported | In the past 12 months, did you seriously think about committing suicide or taking your own life? |

*(Continued)*

**Table 1.** (Continued)

| STUDY | | SAMPLE | | | | OUTCOME | | |
|---|---|---|---|---|---|---|---|---|
| Study and Location | Study Design | Military Sample Size | Demographic and Military Characteristics | Male (%) | Age | Outcome Type | Outcome Definitions | Outcome Measures |
| **Richardson et al (2018)** [65] *Canada* | Retrospective cohort study | 663 | Serving (117) and ex-serving (n = 546) Canadian Armed Forces personnel Treatment-seeking, presented at specialised outpatient mental health clinic for military service related psychiatric conditions (April 2004 to September 2014) Deployed (n = 378) and non-deployed (n = 285) | 91.0% | *Mean age*: 44.6 years | Suicidal ideation | Not reported | In the last 2 weeks have you been bothered by thoughts that you would be better off dead, or hurting yourself in some way? (not at all to nearly every day) [**PHQ-9**] |
| **Sareen et al (2007)**[+2] [46] *Canada* | Cohort study | 8,441 | Serving Canadian Armed Forces personnel–Regular (70.9%) and Reserves (29.1%) Various military ranks including junior NCM (59.8%), senior NCM (20.1%) and officer (20.1%) | 85.3% | 16 to 54 years | Suicidality (suicidal ideation and suicide attempts) | Not reported | **Suicidal Ideation:** In the past 12 months, did you seriously think about committing suicide or taking your own life? **Suicide Attempt:** In the past 12 months, did you attempt suicide or try to take your own life? |
| **Sareen et al (2017)**[+1] [47] *Canada* | Cohort study | 8,161 | Serving Canadian Armed Forces personnel–Regular (n = 6,692) and Reserves (n = 1,469) Various military ranks including junior NCM, senior NCM and officer Deployed (46% of Regulars and 100% of Reservists) and non-deployed Afghanistan era | 86% | 18 to 60 years | Suicidal ideation, suicide plan and suicide attempts [past-year and lifetime] | Not reported | **Suicidal Ideation:** Have you seriously thought about committing suicide or taking your own life? **Suicide Plan:** Have you made a plan for committing suicide? **Suicide Attempt:** Have you attempted suicide or tried to take your own life |
| **Syed Sherriff et al (2019)** [48] *Australia* | Cohort study | **Phase 1:** 24,481 **Phase 2:** 1,798 | Serving Australian Defence Force personnel—Regular only Various military ranks including other ranks (31.64%), non-commissioned officer (NCO; 45.17%) and commissioned officer (21.99%) Combat exposure (31.74%) and deployment experience (63.33%) | 100% | 18 to 60 years | Suicidality (thoughts, plans and attempts) | Not reported | **Phase 1:** **Suicidal Ideation:** In the last 12 months, have you ever felt so low that you thought about committing suicide? **Suicide Plan:** In the last 12 months, have you made a suicide plan? **Suicide Attempt:** In the last 12 months, have you attempted suicide? **Phase 2 [CIDI]:** *Consider each of the following experiences*: **Suicidal Ideation:** Experience A–seriously thought about committing suicide **Suicide Plan:** Experience B–made a plan for committing suicide **Suicide Plan:** Experience C–attempted suicide *If yes, how old were you the first time it happened and has it happened in past 12 months If no to Experience A, B and C were not asked* |

*(Continued)*

**Table 1.** (Continued)

| STUDY | | SAMPLE | | | | OUTCOME | | |
| --- | --- | --- | --- | --- | --- | --- | --- | --- |
| Study and Location | Study Design | Military Sample Size | Demographic and Military Characteristics | Male (%) | Age | Outcome Type | Outcome Definitions | Outcome Measures |
| **Syed Sherriff et al (2020)** [49] *Australia* | Cohort study | **Phase 1:** 3,646 **Phase 2:** 901 | Ex-serving Australian Defence Force personnel—Regular only Left service between 2010 and 2014 Various military ranks including other ranks (52.7%), NCO (31.8%) and commissioned officer (15.6%) Previously deployed (78.9%) | 100% | *With suicidality*: <25 years = 10.8% 26–35 years = 40.3% 36–45 years = 21.9% 46–55 years = 15.8% 56 + years = 11.2% | Suicidality (thoughts, plans and attempts) | Not reported | **Phase 1: Suicidal Ideation:** In the last 12 months, have you ever felt so low that you thought about committing suicide? **Suicide Plan:** In the last 12 months, have you made a suicide plan? **Suicide Attempt:** In the last 12 months, have you attempted suicide? |
| **Taillieu et al (2022)[+1]** [50] *Canada* | Cohort study | 6,692 | Serving Canadian Armed Forces personnel—Regular only Various military ranks including junior NCM (55.1%), senior NCM (24.1%) and officer (20.9%) Deployed (61.5%) and non-deployed (38.5%) | 86.1% | 18 to 60 years | Suicidal ideation and suicide plans [past-year] | Not reported | **Suicidal Ideation:** In the past 12 months, have you seriously thought about committing suicide or taking your own life? **Suicide Plan:** In the past 12 months, have you made a plan for committing suicide? |
| **Thompson et al (2014)** [51] *Canada* | Cohort study | 2,658 | Ex-serving Canadian Armed Forces personnel—Regular only Released from service between January 1998 and December 2007 Various military ranks including recruits (n = 249), junior NCM (n = 1,049), senior NCM (n = 880) and officer/ cadets (n = 480) Deployed and non-deployed | 89% | *Mean age:* 43.5 years | Suicidal ideation | Not reported | Have you seriously thought about committing suicide or taking your own life? *If yes*: Has this happened in the past 12 months? |
| **Thompson et al (2019)** [52] *Canada* | Cohort study | 2,755 **Recently released:** 1,180 **Earlier released:** 1,575 | Ex-serving Canadian Armed Forces—Regular only 'Recently released' between September 2012 to 2015 'Earlier released' between 1998 to August 2012 | 87.2% | *Mean age:* **Recently released:** 41 years **Earlier released:** 50 years | Suicidal ideation [past-year] | Not reported | Have you seriously thought about committing suicide or taking your own life? *If yes*: Has this happened in the past 12 months? |
| **VanTil et al (2021)** [66] *Canada* | Retrospective cohort study | 220,734 | Ex-serving Canadian Armed Forces—Regular (95%) and Reserves (5%) Released from service between 1976 and 2012 Various military ranks including junior/senior NCM (82%) and officer (18%) | 89% | *Age at release:* <25 to >45 years | Completed suicide | ICD-8 and ICD-9 codes E950-E959 (*suicide and self-inflicted injury*) and ICD-10 codes X60-X84 (*intentional self-harm*) and Y87.0 (*sequelae of intentional self-harm*) | ICD-8 and ICD-9 codes E950-E959 (*suicide and self-inflicted injury*) and ICD-10 codes X60-X84 (*intentional self-harm*) and Y87.0 (*sequelae of intentional self-harm*) |
| **Varker et al (2022)** [55] *Australia* | Cohort study | 12,806 **Serving:** 8,480 **Ex-serving:** 4,326 | Serving and ex-serving Australian Defence Force personnel—Regular and Reserves If ex-serving, left Australian Defence Force in last 5 years | Male and female included but % not reported | 18 to 58+ years | Suicidality (suicidal ideation and plans, and suicide attempts) | Not reported | In the last 12-months, have you: *Felt that life was not worth living *Felt so low that you thought about committing suicide *Made a suicide plan *Attempted suicide **[National Survey of Mental Health and Wellbeing/Adult Psychiatric Morbidity Survey]** |

*(Continued)*

**Table 1.** (Continued)

| STUDY | | SAMPLE | | | | OUTCOME | | |
|---|---|---|---|---|---|---|---|---|
| Study and Location | Study Design | Military Sample Size | Demographic and Military Characteristics | Male (%) | Age | Outcome Type | Outcome Definitions | Outcome Measures |
| **Williamson et al (2021)** [53] *UK* | Cohort study | 402 | Ex-serving UK Armed Forces Personnel Various military ranks including junior rank, NCO and officer Deployed and non-deployed | 88.6% | *Mean age*: **Moral injury:** 50.0 years **Trauma:** 50.4 years **Mixed:** 51.3 years **No event:** 52.1 years | Suicidal ideation | Not reported | Suicidal behaviors questionnaire-revised [**SBQ-R**] |
| **Woodhead et al (2011)** [54] *UK* | Cohort study | 257 | Ex-serving UK Armed Forces personnel Post-national service era | 85.2% | 16 to 64 years *Median age*: **Post-national service:** Male: 49 years Female: 45 years | Self-harm (suicidal thoughts, suicide attempts and self-harm) | Not reported | **Suicidal Ideation:** Have you ever thought of taking your own life **Suicide Attempt:** Have you ever made an attempt to take your life **Self-Harm:** Have you ever deliberately harmed yourself in any way but without the intention of killing yourself [**Revised Clinical Interview Schedule**] |

**Note.** [+1] use data from the Canadian Forces Mental Health Survey; [+2] use data from the Canadian Community Health Survey: Mental Health and Well-Being Canadian Forces Supplement (CCHS-CFS); [+3] use data from the Scottish Veterans Health Study; [+4] use data from various phases of the King's Centre for Military Health Research (KCMHR) Health and Wellbeing Study; NCO: Non-commissioned officer; NCM: Non-commissioned member; ICD: International Classification of Diseases; NHS: National Health Service (UK).

### 3.4. Definition and measurement of self-harm and suicide behaviour outcomes

Only six papers provided a definition for the outcomes of interest (**Table 1**), with the remaining papers failing to report why these terms were not defined.

The majority of papers used self-report surveys (n = 17) [15,29,43–47,50,53,55–58,60,61,65]. Two papers collected data via structured clinical interviews [51,52] and four utilised a mix of self-report surveys and structured clinical interviews [18,28,48,49]. One study employed clinical assessment [59]. The remaining papers used International Classification of Diseases (ICD) codes to define and measure the outcome of interest [62–64,66].

A variety of outcome measures were used to collect data on the outcomes of self-harm and suicidal behaviours (**Table 1**). For instance, ICD codes [62–64,66], Patient Health Questionnaire-9 [43,65], Clinical Interview Schedule-Revised [15,18] and Suicidal Behaviors Questionnaire-Revised [53].

### 3.5. Risk factors

Numerous risk and protective factors were identified in the review with varying levels of statistical significance (**Table 2**).

**Table 2. Risk and Protective Factors Identified in Included Studies (N = 28).**

| Study and Location | Outcome | Risk factors identified | | Protective factors identified | |
|---|---|---|---|---|---|
| | | Significant | Non-Significant | Significant | Non-Significant |
| **Afifi et al (2016)** [56] *Canada* | Suicidal ideation, suicide plans and suicide attempts | Suicidal Ideation:<br>• Child abuse exposure<br>Deployment-related trauma [non-significant after full adjustment]<br>Child abuse exposure with/without deployment-related trauma [Regulars]<br>Experiencing child abuse exposure only, relative to deployment-related trauma only [Regulars]<br>Suicide Plans:<br>• Child abuse exposure<br>• Deployment-related trauma<br>Child abuse exposure with/without deployment-related trauma [Regulars]<br>Suicide Attempts:<br>• Child abuse exposure | Suicidal Ideation:<br>• • Deployment-related trauma without history of child abuse exposure [Regulars]<br>Suicide Plans:<br>• • Deployment-related trauma without history of child abuse exposure [Regulars] | - | - |
| **Belik et al (2009)** [57] *Canada* | Suicide attempts | Men:<br>• Sexual and other interpersonal traumas<br>• Increasing number of traumatic events<br>• Deployment-related traumatic events:<br> *Purposely injured, tortured, or killed someone associated<br> *Witnessing atrocities [non-significant after full adjustment]<br>• Accident or other unexpected trauma:<br> *Toxic chemical exposure<br> *Other accident [non-significant after full adjustment]<br> *Life threatening illness<br> *Caused death accidently [non-significant after full adjustment]<br>• Civilian in religious terror<br>• Event happened to other<br>• Other trauma<br>Women:<br>Sexual and other interpersonal traumas [Being mugged/kidnapped non-significant after full adjustment]<br>Accident or other unexpected trauma:<br> *Other accident [non-significant after full adjustment]<br> *Man-made disaster<br> *Unexpected death [non-significant after full adjustment]<br> *Witness death [non-significant after adjustment]<br>Event happened to other | Men:<br>• Exposure to combat operations<br>• Accident or other unexpected trauma:<br> *Auto accident<br> *Man-made disaster<br> *Natural disaster<br> *Unexpected death<br> *Child illness or injury<br> *Witness death<br>• Civilian in war zone<br>Women:<br>• Exposure to combat or peacekeeping operations<br>• Accident or other unexpected trauma:<br> *Auto accident<br> *Toxic chemical exposure<br> *Child illness or injury<br> *Life threatening illness<br>• Civilian in war zone<br>• Other trauma | Women:<br>Civilian in religious terror | Men:<br>Exposure to peacekeeping operations |
| **Bergman et al (2019)*** [62] *UK* | Self-harm | Veterans [vs non-veterans]<br>• Sex: Male<br>• Age: Oldest (1945–1949) and youngest (1980–1985) cohorts<br>• Early service leavers (ESLs; ≤2.5 years service), especially those who did not complete initial training<br>• Older veterans with 4–6 years and 7–9 years service<br>Comorbid disorders: PTSD, anxiety, lung cancer, diabetes [veterans vs non-veterans] | • Sex: Female<br>Comorbid disorders: Any cancer, alcoholic liver disease, mood disorder [veterans vs non-veterans] | • Length of service: 23+ years of service<br>• Age: Middle-aged (born 1960 onwards) | • Length of service: Longer 10–12 years, 13–16 years, 17–22 years |
| **Bergman et al (2022)*** [63] *UK* | Completed suicide | • Older women veterans (>40 years)<br>• Age: 1950–1954 cohort<br>• ESLs (≤3 years service) | • Sex: Female<br>• Age: Most recent birth cohort (1985–1995)<br>• Length of service: 17–22 years<br>• Health: History of mood disorder, PTSD, history of cardiovascular disease | - | • Male veterans vs male non-veterans<br>Length of service: >22 years [remained non-significant after adjustment]<br>• Health: Diagnosis of alcoholic liver disease |

*(Continued)*

**Table 2.** (Continued)

| Study and Location | Outcome | Risk factors identified | | Protective factors identified | |
|---|---|---|---|---|---|
| | | Significant | Non-Significant | Significant | Non-Significant |
| **Boulos & Fikretoglu (2018) [58]** *Canada* | Suicidal ideation | Reserves vs Regulars *[non-significant after adjustment]* • Deployment: Shorter intervals (<4 years) since return from Afghanistan deployment • Deployment-related experiences: *Ever known someone seriously injured or killed *Ever found yourself in life-threatening situation where you were unable to respond because of the rules of engagement *Ever been injured *Ever felt responsible for death of Canadian or ally personnel | • Deployment: Higher total number of Afghanistan deployments (i.e., 2 + deployments) | • Deployment: *Higher cumulative duration of Afghanistan-related deployments (≥361 days) *Less time away on deployment in past 3 years (<6 months, 7–12 months, 13–24 months) | Deployment: *Afghanistan deployment location (Kabul, Kandahar, Multiple) *Other non-Afghanistan deployments *Moderate cumulative duration of Afghanistan-related deployments (121–140 days, 241–360 days) Deployment-related experiences: *Ever seen ill or injured women or children who you were unable to help *Ever received incoming artillery, rocket or mortar fire *Ever had a close call for example shot or hit but protective gear saved you |
| **Harden & Murphy (2018) [59]** *UK* | Suicidal ideation | • ESLs (<4 years service) • Employment status: Unemployed • Childhood adversity: High pre-service childhood adversity | • Sex: Female • Relationship status: Not in a relationship • Combat role • Non-voluntary discharge from service | • Age: Middle age groups (35–44 years, 45–54 years) • Time to seek help: Longer than 5 years | • Age: Older age groups (55+ years) • Financial difficulties |
| **Hines et al (2013) [60]** *UK* | Self-harm | • Age: Younger age • Sex: Female • Marital status: Separated/divorced Ex-serving *[vs serving]* • Social support: *Less number of close family/friends (None, 1–2 close) *More family relationship adversity (2+ items) • Childhood adversity: *Spent time in local authority care *Childhood anti-social behaviour *[only presented at unadjusted level]* | - | Reserves *[vs Regulars]* • Age: Older age • Higher number of social activities | • Higher number of close family/friends (6–10 close, 11–15 close) • Deployment experience |
| **Jones et al (2019) [18]** *UK* | Self-harm, suicidal ideation and suicide attempts | Self-Harm: Ex-serving *[vs serving]* • Sex: Female • Health: Probable diagnosis of depression, probable diagnosis of anxiety disorder, probable diagnosis of PTSD, lifetime suicidal ideation • Mental health stigmatisation (Moderate score 11–17, Higher score 18–24) • Perceived practical barriers to care (Higher score 6–11) Negative attitudes to mental illness (Moderate score 11–12, Higher score 13–23) *[Higher score non-significant after full adjustment]* Help-seeking: Formal medical *[non-significant after full adjustment]* Suicide Attempts: Reserves *[vs Regulars]* • Health: Probable diagnosis of depression, probable diagnosis of anxiety disorder, probable diagnosis of PTSD, lifetime suicidal ideation • Perceived practical barriers to care: Higher score (6–11) • Negative attitudes to mental illness: Moderate score (11–12), higher score (13–23) • Help-seeking: Formal medical | Self-Harm: • Health: Probable alcohol misuse Suicide Attempts: Ex-serving *[vs serving]* • Mental health stigmatisation: Moderate score (11–17), higher score (18–24) • Help-seeking: Informal | Self-Harm: Age: Middle and older age groups (e.g. 35–39 years, 40–49 years, >49 years) *[significant at varying levels of adjustment]* Rank: More senior rank (Junior NCO to warrant officer, commissioned officer) *[significant at varying levels of adjustment]* • Perceived social support (Higher score 33–36) Suicide Attempts: • Rank: More senior rank (Commissioned officer) • Service branch: RAF • Perceived social support: Higher score (33–36) | Self-Harm: Reserves *[vs Regulars]* • Service Branch: RAF Suicide Attempts: • Rank: NCO to warrant officer • Health: Probable alcohol misuse • Help-seeking: Formal non-medical |
| **Jones et al (2021) [61]** *UK* | Self-harm | • Junior entrants enlisted between April 2003 and March 2013 | • Junior entrants <17.5 years | - | - |
| **Kapur et al (2009) [64]** *UK* | Completed suicide | • Age: Younger age at discharge (16–19 years, 20–24 years) Training status: Untrained *[non-significant after adjustment]* • Length of service: Shorter (<1 year, 2 to <3 years, 3 to <4 years) | • Type of discharge: Medical • Length of service: Moderate (5 to <10 years) | • Sex: Female • Marital status: Married, unknown • Service branch: Naval Service, RAF • Rank: More senior rank (Officer) | • Age: Older age at discharge (40–44 years) • Length of service: Moderate (4 to <5 years), longer (10 to <15 years) |

*(Continued)*

**Table 2.** (Continued)

| Study and Location | Outcome | Risk factors identified | | Protective factors identified | |
|---|---|---|---|---|---|
| | | Significant | Non-Significant | Significant | Non-Significant |
| **Kerr et al (2018)*** [29] *Australia* | Suicide attempts | • Employment status: Not working/totally or permanently incapacitated<br>• Health: PTSD symptom severity | • Employment status: Retired, working part-time<br>• Conflicts served: Both periods (pre-East Timor and from East Timor onwards)<br>Health: Anxiety, depression | • Employment status: Employed | Deployment [vs no deployment]<br>• Conflicts served: Pre-East Timor, from East Timor onwards |
| **Ketcheson et al (2018)** [43] *Canada* | Suicidal ideation | • Age: Older age<br>• Length of service: Shorter<br>• Lower levels of perceived social support (Low, medium) | - | - | - |
| **Nelson et al (2011)** [44] *Canada* | Suicidal ideation | • Higher number of lifetime traumatic experiences (including exposure to combat)<br>• Health: Past year diagnosis of PTSD or major depressive disorder (MDD)<br>• Past-year MDD mediates relationship between number of lifetime traumatic events and past-year SI | - | • Higher levels of perceived social support | - |
| **O'Toole et al (2015)*** [28] *Australia* | Lifetime suicidality | Suicidal ideation:<br>Health: PTSD, alcohol dependence, cannabis abuse, depression single (moderate and severe), depression recurrent (mild, moderate, severe), depression lifetime, generalised anxiety disorder, obsessive compulsive disorder, panic without agoraphobia, social phobia, situational phobia<br>Suicide plan:<br>• Health: PTSD, alcohol dependence, cannabis abuse, depression single (moderate and severe), depression recurrent (moderate, severe), depression lifetime, generalised anxiety disorder, panic without agoraphobia, situational phobia<br>Suicide attempt:<br>• Health: PTSD, alcohol dependence, depression recurrent (mild, moderate) depression lifetime, agoraphobia without panic, panic with agoraphobia, social phobia, situational phobia | Suicidal ideation:<br>• Health: Depression single (mild), dysthymia, agoraphobia without panic, panic with agoraphobia<br>Suicide plan:<br>• Health: Depression single (mild), depression recurrent (mild), obsessive compulsive disorder, agoraphobia without panic, panic with agoraphobia, social phobia<br>Suicide attempt:<br>• Health: Depression single (mild, moderate, severe), depression recurrent (moderate), dysthymia, generalised anxiety disorder, obsessive compulsive disorder | - | Suicidal ideation:<br>Health: Alcohol abuse<br>Suicide plan:<br>Health: Alcohol abuse, dysthymia<br>Suicide attempt:<br>Health: Alcohol abuse, cannabis abuse |
| **Pinder et al (2012)** [15] *UK* | Self-harm | • Age: Younger age<br>Ex-serving [vs serving]<br>• Childhood adversity: Higher<br>• Health: PHQ or PTSD diagnosis, any PHQ diagnosis, any depressive syndrome, somatization disorder, PTSD | • Education: Low educational attainment (No qualifications)<br>Marital status: Single/Not in a relationship, divorced/separated/widowed [non-significant at unadjusted level]<br>• Service branch: Army, RAF<br>• Rank: More junior rank<br>Length of service: Shorter [significant at unadjusted level]<br>Deployment: Deployment to Iraq [non-significant at unadjusted level]<br>• Heath: Any anxiety syndrome, alcohol abuse | Rank: More senior rank (Officer) [non-significant after adjustment]<br>Length of service: Longer term<br>Childhood adversity: Less (0–1 factors, 2–3 factors) | • Age: Older age<br>Sex: Female [non-significant at unadjusted level]<br>Education: Higher educational attainment (A-Levels, degree) [Degree significant at unadjusted level]<br>Reserve status [significant at unadjusted level]<br>Rank: More senior rank (Officer) [significant at unadjusted level]<br>Service branch: Navy (including Marines) [significant at unadjusted level]<br>• Childhood adversity: Less (4–5 factors) |
| **Richardson et al (2017)*** [45] *Canada* | Suicidal ideation | • Health: PTSD, MDD, generalised anxiety disorder, panic disorder, alcohol abuse and dependency, insomnia (incremental continuous change)<br>• Past-year mental health status (1 disorder, 2 + disorders)<br>• Marital status: Divorced/separated/widowed | • Age: Younger age groups (30–39 years)<br>• Marital status: Single/never married<br>• No deployments to Afghanistan | - | • Age: Middle/older age groups (40–49 years, 50–60 years)<br>• Language: French |
| **Richardson et al (2018)** [65] *Canada* | Suicidal ideation | • Health: Depressive symptom severity<br>• Depressive symptom severity mediates relationship between both sleep disturbances and nightmares, with suicidal ideation | • Sleep disturbances<br>• Nightmares<br>• Health: PTSD symptom severity, anxiety symptom severity | - | - |
| **Sareen (2007)** [46] *Canada* | Suicidality (suicidal ideation and suicide attempts) | • Deployment experiences:<br>   *Witnessing atrocities or massacres<br>Perceived need for mental healthcare and no help seeking or DSM diagnosis [significant at unadjusted level] | • Deployment experiences: Exposure to combat | - | • Deployment experiences: Exposure to peacekeeping operations |

(*Continued*)

**Table 2.** (Continued)

| Study and Location | Outcome | Risk factors identified | | Protective factors identified | |
|---|---|---|---|---|---|
| | | Significant | Non-Significant | Significant | Non-Significant |
| **Sareen et al (2017)** [47] *Canada* | Suicidal ideation, suicide plan and suicide attempts [past-year and lifetime] | Suicidal Ideation (past-year): Deployment experiences: *[significant at varying levels of adjustment]* *Combat exposure (lifetime) *Peacekeeping (lifetime) *Witnessed atrocities (lifetime) *Known someone injured or killed (CF deployment) *Ever in life-threatening situation and unable to respond due to rules of engagement (CF deployment) *Ever been injured (CF deployment) *Seen ill or injured women or children (CF deployment) *Received incoming artillery, rocket, or mortar fire (CF deployment) *Felt responsible for death of Canadian or ally personnel (CF deployment) *Had a close call but protective gear saved you (CF deployment) *Had difficulty distinguishing combatants from non-combatants (CF deployment) • Total number deployment-related traumatic events (DRTEs) Suicide Plan (past-year): Deployment experiences: *[significant at varying levels of adjustment]* *Peacekeeping (lifetime) *Witnessed atrocities (lifetime) *Known someone injured or killed (CF deployment) *Ever in life-threatening situation and unable to respond due to rules of engagement (CF deployment) *Ever been injured (CF deployment) *Seen ill or injured women or children (CF deployment) *Received incoming artillery, rocket, or mortar fire (CF deployment) *Felt responsible for death of Canadian or ally personnel (CF deployment) *Had a close call but protective gear saved you (CF deployment) *Had difficulty distinguishing combatants from non-combatants (CF deployment) Total number DRTEs *[none remained significant after full adjustment]* Suicide Attempt (past-year): Deployment experiences: *[significant at varying levels of adjustment]* *Combat exposure (lifetime) *Peacekeeping (lifetime) *Witnessed atrocities (lifetime) *Ever in life-threatening situation and unable to respond due to rules of engagement (CF deployment) *Ever been injured (CF deployment) *Had a close call but protective gear saved you (CF deployment) *Had difficulty distinguishing combatants from non-combatants (CF deployment) Total number DRTEs | Suicidal Ideation (past-year): • Deployment experiences: *Military sexual trauma during deployment • Lifetime deployment Suicide Plan (past-year): • Deployment experiences: *Combat exposure, (lifetime) • Lifetime deployment Suicide Attempt (past-year): • Deployment experiences: *Known someone injured or killed *Seen ill or injured women or children (CF deployment) *Received incoming artillery, rocket, or mortar fire (CF deployment) • Lifetime deployment | - | - |
| **Syed Sherriff et al (2019)** [48] *Australia* | Suicidality (thoughts, plans and attempts) | • Education: Lower educational attainment (Year 10 or below) • Age: Several age groups (e.g. 25–34 years, 35–44 years, 45+ years) • Relationship status: Single • Childhood trauma: High counts (3+), interpersonal (without non-interpersonal), both interpersonal and non-interpersonal • Childhood disorder: Anxiety • Adult trauma: High counts (3+), non-interpersonal • Adult-onset disorder: Depression • Rank: More junior ranks (NCOs, other ranks) | • Childhood trauma: Low counts (1–2), unclassified, non-interpersonal (without interpersonal) • Childhood disorder: Depression, alcohol • Adult trauma: Low counts (1–2), non-intimate interpersonal • Adult-onset disorder: Anxiety, alcohol • Combat exposure | • Education: Higher educational attainment (Certificate or diploma, Year 11/12, university degree) • Relationship status: Current significant relationship • Service branch: RAF | • Adult trauma: Intimate interpersonal • Service branch: Army • Deployment: Previously deployed |

(*Continued*)

**Table 2.** (Continued)

| Study and Location | Outcome | Risk factors identified | | Protective factors identified | |
|---|---|---|---|---|---|
| | | **Significant** | **Non-Significant** | **Significant** | **Non-Significant** |
| **Syed Sherriff et al (2020)** [49] *Australia* | Suicidality (thoughts, plans and attempts) | • Rank: More junior ranks (NCO, other)<br>• Deployment: Previously deployed<br>Childhood-onset trauma: Interpersonal *[remains significant after full adjustment]*<br>• Adult-onset trauma: Mean number of types<br>Childhood-onset disorder: Anxiety, any disorder *[Anxiety remains significant after full adjustment]*<br>Adult-onset disorder: Anxiety, depression, single, multiple *[Anxiety and depression remain significant after full adjustment]*<br>Previously Deployed:<br>• Rank: More junior ranks (NCO, other)<br>Childhood-onset trauma: Interpersonal *[remains significant after full adjustment]*<br>Childhood-onset disorder: Anxiety *[remains significant after full adjustment]*<br>• Adult-onset disorder: Anxiety, depression, alcohol, single, multiple<br>Deployment trauma exposure types *[remains significant after full adjustment]*<br>• Non-deployment adult trauma: Multiple, mean | • Childhood-onset trauma: Non-interpersonal, 1–2 traumas, 3+ traumas, mean number of types<br>• Adult-onset trauma: Combat<br>• Childhood-onset disorder: Depression, alcohol<br>• Adult-onset disorder: Alcohol<br>Previously Deployed:<br>• Childhood-onset trauma: Non-interpersonal, 1–2 traumas, 3+ traumas)<br>• Childhood-onset disorder: Alcohol, any disorder<br>• Combat exposure<br>• Non-deployment adult trauma: Non-interpersonal, interpersonal | • Relationship status: Current relationship<br>• Educational attainment: Higher (Year 11/12, university degree)<br>Previously Deployed:<br>• Relationship status: Current relationship | • Educational attainment: Higher (Certificate or diploma)<br>• Service branch: Navy, Air Force<br>• Childhood-onset trauma: Other<br>Previously Deployed:<br>• Age: Several age groups (e.g. 26–35 years, 36–45 years, 46–55 years, 56 + years)<br>• Educational attainment: Higher (Year 11/12, certificate or diploma, university degree)<br>• Service branch (Navy, Airforce)<br>• Childhood-onset trauma: Other<br>• Childhood-onset disorder: Depression<br>• Non-deployment adult trauma: Single trauma |
| **Taillieu et al (2022)** [50] *Canada* | Suicidal ideation and suicide plans (past-year) | Suicidal Ideation:<br>• History of child abuse<br>• Health: MDD, generalised anxiety disorder, panic attacks, PTSD<br>Suicide Plans:<br>History of child abuse *[non-significant after full adjustment]*<br>Exposure to DRTEs *[non-significant after full adjustment]*<br>• Health: MDD, PTSD, alcohol abuse or dependence | Suicidal Ideation:<br>• Exposure to DRTEs<br>Health: Alcohol abuse or dependence<br>Suicide Plans:<br>• Health: Generalised anxiety disorder, panic disorder, panic attacks | - | Suicidal Ideation:<br>• Health: Panic disorder |
| **Thompson et al (2014)** [51] *Canada* | Suicidal ideation | • Physical health conditions: Gastrointestinal condition<br>• Mental health conditions: Depression or anxiety, mood disorders<br>• Number of physical health conditions (0–8)<br>• Number of mental health conditions (0–4)<br>• Marital status: Single never married, separated/divorced/widowed | Physical health conditions: Chronic pain or discomfort, diabetes, respiratory condition, cardiovascular condition, obesity, hearing problem *[all significant at unadjusted level]*<br>Mental health conditions: Anxiety disorder, PTSD *[both significant at unadjusted level]*<br>Age: Younger and middle age groups (20–34 years, 35–49 years) *[35–49 years group significant at unadjusted level]*<br>Sex: Female *[significant at unadjusted level]*<br>Educational attainment: High school graduation, some post-secondary education *[both significant at unadjusted level]*<br>Income: Below low income measure *[significant at unadjusted level]*<br>Rank: Junior ranks (Recruits, junior NCM, senior NCM) *[Junior and senior NCMs significant at unadjusted level]* | SF-12 physical health<br>SF-12 mental health | - |
| **Thompson et al (2019)** [52] *Canada* | Suicidal ideation (past-year) | Perceived adjustment: Neither, difficult *[significant at unadjusted level]*<br>• Group identity:<br>   *Strong sense of local/community belonging but not part of a group<br>   *Weak sense of local/community belonging but part of a group<br>   *Weak sense of local/community belonging but not part of a group<br>• Mental health problem: Mild/moderate, Severe | Age: Middle and older age groups (30–39 years, 40–49 years, 50–59 years, 60 + years) *[40–49 years, 50–59 years age groups significant at unadjusted level]*<br>Marital status: Single/never married, widowed/separated/divorced *[latter significant at unadjusted level]*<br>Main employment-related activity year prior: Looked for work, school or training, retired/not looking, disabled *[looked for work, school or training, disabled significant at unadjusted level]*<br>Household income: Lower/Moderate (Quintiles 1–4) *[Quintiles 1–3 significant at unadjusted level]*<br>Physical conditions: More (1 condition, 2 conditions, 3+ conditions) *[2 conditions, 3 + conditions significant at unadjusted level]* | - | • Age: Younger age groups (<30 years) |

*(Continued)*

**Table 2.** (Continued)

| Study and Location | Outcome | Risk factors identified | | Protective factors identified | |
|---|---|---|---|---|---|
| | | Significant | Non-Significant | Significant | Non-Significant |
| **VanTil et al (2021)** [66] *Canada* | Completed suicide | Male:<br>• Age at release: Younger and middle-age groups (<25 years, 25–34 years, 35–44 years)<br>• Rank at release: More junior rank (Junior NCM)<br>Female:<br>• Rank at release: More junior rank (NCM) | Female:<br>• Age at release: Younger age groups (<25 years)<br>• Reserve status | - | Male:<br>Rank at release: More senior rank (Senior officer) *[all significant at unadjusted level]*<br>Reserve status *[significant risk factor at unadjusted level]* |
| **Varker et al (2022)** [55] *Australia* | Suicidality (suicidal ideation and plans, and suicide attempts) (past-year) | Ex-serving:<br>• Health:<br> *DAR-5 anger total score<br> *PCL-5 PTSD total score<br> *PHQ-9 depression total score<br> *AUDIT alcohol use total score<br>• Relationship status: Not in a relationship, In a relationship but not living together<br>• Type of discharge: Medical discharge | Ex-serving:<br>• Service branch: Navy, Air Force | - | Ex-serving:<br>• Sex: Male<br>• Ex-serving status |
| **Williamson et al (2021)** [53] *UK* | Suicidal ideation | • Exposure to morally injurious events<br>• Exposure to traumatic events<br>• Exposure to mixed events | - | - | - |
| **Woodhead et al (2011)** [54] *UK* | Self-harm (suicidal thoughts, suicide attempts and self-harm) | Self-Harm (ever):<br>• ESLs (<4 years service)<br>Suicidal Thoughts (ever):<br>• ESLs (<4 years service)<br>Ex-serving female *[vs non-veterans]* | Self-Harm (ever):<br>Ex-serving female *[vs non-veterans]*<br>Suicidal Thoughts (ever):<br>Ex-serving male *[vs non-veterans]*<br>Suicide Attempts (ever):<br>• ESLs (<4 years service)<br>Ex-serving female and ex-serving male *[vs non-veterans]* | - | Self-harm (ever):<br>Ex-serving male *[vs non-veterans]* |

**Note.** Significance refers to adjusted effect sizes where available. If only unadjusted effect sizes are presented, studies have been marked by an asterisk (*).

ESL: Early service leaver; DRTE: Deployment-related traumatic event; RAF: Royal Air Force; CF: Canadian Forces; NCO: Non-commissioned officer; NCM: Non-commissioned member; DSM: Diagnostic and Statistical Manual of Mental Disorders; PTSD: Post-traumatic stress disorder; MDD: Major depressive disorder; DAR-5: Dimensions of Anger Reactions 5-item; PCL-5: PTSD Checklist for DSM-5; PHQ-9; Patient Health Questionnaire; AUDIT: Alcohol Use Disorder Identification Test.

**3.5.1. Risk factors for self-harm.** Pre-enlistment factors increased the risk of self-harm, including experiencing childhood adversity and abuse than those without these experiences (approximately doubling the risk) [15,60]. Certain demographic groups were also at heightened risk, including younger age groups [15,60,61] compared to middle/older age groups, and those no longer in a relationship (i.e., separated/divorced) [60] compared to in a relationship. There were discrepancies between studies regarding the impact of sex, with two studies reporting the risk of self-harm was increased for females [18,60] but one study stated increased risk for males [62]. Military characteristics associated with increased risk of self-harm included having left service, with ex-serving personnel reported to be as much as three times more at risk of self-harm than serving personnel or non-military population depending on sample explored [15,18,60,62]. Other military characteristics that increased the risk of self-harm included being an early service leaver, defined within the relevant papers as <4 years [54] and ≤2.5 years [62] of service. This was also apparent for those with a shorter length of service (but not an early service leaver), for instance 4 to 6 years of service [62].

Several health-related factors also increased risk of self-harm. The factors with some of the largest effect sizes (ranging from approximately two to eight times more likely) included clinical or probable diagnosis of PTSD [15,18,62], depression [15,18], and anxiety [18,62], as well as history of suicidal ideation [18]. Other risk factors for self-harm included a lack of social support [60], mental health-related stigmatisation [18], perceived practical barriers to care

[18], negative attitudes to mental illness [18], and formal medical help-seeking [18] (all approximately tripling risk).

**3.5.2. Risk factors for suicide behaviours.** Similar risk factors were identified for suicide behaviours (i.e., suicidal ideation, suicide attempts and completed suicide). Pre-enlistment factors, such as experience of childhood adversity and abuse were found to increase the risk of suicide compared to those without these experiences (ranging from two to seven times more likely) [48–50,56,59]. Certain demographic characteristics increased the risk of suicide behaviours, including being single or no longer in a relationship (i.e., separated, divorced, widowed) [45,48,51,55], lower educational attainment (i.e., less than university degree level) [48], and unemployment [29,59]. Another identified risk factor for suicide behaviours was a lack of social support [43]. The influence of age remains unclear, as studies reported a variety of age groups to be at increased risk of suicide behaviours [43,48,63,64,66]; for instance younger age groups (e.g., 16 to 24 years [64]), younger/middle age groups (e.g., <25 to 44 years [66]), and a mixture of age groups (i.e., 25 to 45+ years [48]).

Numerous military characteristics increased the risk of suicide behaviours, including being medically discharged from service [55], holding a more junior rank during service [48,49,66] (e.g., non-commissioned officer or other lower ranks [48,49]), serving as a Reservist (vs Regular) [18], and shorter intervals since return from last Afghanistan deployment (i.e., <4 years) [58]. Being an early service leaver was another factor that increased risk of suicide behaviours, defined within the relevant papers as ≤3 years [63] and <4 years [54,59] of service (e.g., increased risk by as much as eight-fold [59]). Although not referred to as early service leavers, having a shorter length of service (i.e. one to four years) was also a risk factor for completed suicide among a sample of UK ex-serving personnel [64]. Experience of deployment-related traumatic events [44,46,47,50,56–58] and exposure to trauma [44,48,49,53,57] were other factors associated with increased risk of suicide behaviours.

Several health-related factors were positively associated with suicide behaviours among serving and ex-serving personnel. These included number of physical health disorders (small effect size of just over one) [51] and number of mental health disorders (odds ranging from around two to 20 times more likely) [45,51,52]. More specifically, clinical or probable diagnosis of PTSD [18,28,29,44,45,50,55], depression [18,28,44,45,48–51,55,65], anxiety [18,28,45,49,50,55], alcohol use disorder [28,45,50,55], cannabis use [28], mood disorder [51], insomnia [45], or panic attacks [50] (ranging from just over one to 15 times more likely). Other risk factors included a higher number of perceived practical barriers to care, negative attitudes to mental illness, and formal medical help-seeking [18]. Further, lifetime suicidal ideation was reported as a risk factor for lifetime suicide attempts among a sample of serving and ex-serving personnel (up to 12 times more likely) [18].

## 3.6. Protective factors

A number of protective factors were also identified (**Table 2**).

**3.6.1. Protective factors for self-harm.** Several factors were identified that decreased the risk of self-harm, including middle/older age groups [18,60,62] (e.g. 35 to 49+ years [18]) compared to other age groups, and experience of less adversity during childhood (i.e., less than three factors) [15]. Certain military characteristics reduced the likelihood of self-harm, for example holding a more senior rank (i.e., officer vs junior ranks) [15,18], being a Reservist (vs Regular) [60], and having a longer length of service [15,62] (e.g., ≥23 years [62]). Further, a higher level of perceived social support was associated with a lower likelihood of self-harm [18,60] (e.g., a score of 33–36 on the multidimensional scale of perceived social support [18]).

**3.6.2. Protective factors for suicide behaviours.** When considering suicide behaviours (i.e., suicidal ideation, suicide attempts and completed suicide), several identified factors reduced risk. Certain demographics were significantly associated with decreased risk of suicide behaviours including being female [64], middle age groups (e.g. 35 to 54 years compared to those <35 years) [59], having a current significant relationship (i.e., being married or in a relationship; approximately half as likely than those no longer in relationships) [48,49,64], higher educational attainment (vs lower educational attainment) [48,49] (e.g., higher than Year 10 [48,49]), and being employed (vs unemployed) [29].

Several military characteristics decreased the risk of suicide behaviours, including holding a more senior rank (vs junior ranks) [18,64] (e.g., officer rank [64]), service in the Royal Air Force [18,64] or Naval Services [64] (vs Army), and certain deployment-related factors such as higher cumulative duration of Afghanistan-related deployments (i.e., ≥361 days) [58], and less time away on deployment in the past three years (i.e., up to two years) [58] (all approximately half as likely). Higher levels of perceived social support [18,44] (e.g., a score of 33–36 on the multidimensional scale of perceived social support [18]), and taking longer than five years to seek support (vs less than five years) [59] were also identified factors that reduced the likelihood of suicide behaviours.

## 4. Discussion

This review identified 28 papers reporting on a range of factors associated with self-harm and suicide behaviours among serving and ex-serving personnel of the UK Armed Forces, Canadian Armed Forces, and Australian Defence Force.

A variety of definitions and measurements were used for the outcomes reported in the included papers. Definition is an important aspect of academic research and clinical practice, yet precise definitions have been contested, and current terminology varies across nations [68]. For instance, there are several terms for self-harm, including non-suicidal self-injury, deliberate self-harm and self-inflicted violence. Definitions of mental health conditions and the use of consistent language are an important starting point within good quality research papers and are also important for reducing stigma and encouraging individuals to seek help, particularly relating to self-harm and suicide behaviours [69].

Several of the identified risk factors reflect those among the general population [3,70], serving and ex-serving military personnel of the US Armed Forces [27,33], and other similar occupational groups (such as emergency responders) [71], and were generally consistent across the included geographical regions. Some similar identified risk factors for suicide among the US military community include mental health diagnoses (e.g. mood, alcohol/substance, psychotic and personality disorders), and experience of childhood adversities [27].

Although no relevant peer-reviewed papers were identified from New Zealand, suicide prevention work is being conducted [72]. Risk factors for suicide among the New Zealand Defence Force include current mental health concerns, acute life stressors and negative attitudes towards help-seeking, and protective factors include positive mental health and social support [72]. The paucity of peer-reviewed work from New Zealand may be due to the smaller military population of the New Zealand Defence Force compared to the military populations of other included nations, making large quantitative studies more difficult.

It is important to recognise that some risk and protective factors were not always consistent across included studies in this review, and the influence of age, sex, certain military service characteristics, and certain health diagnoses remains unclear. For instance, having served as a Reservist increased the likelihood of suicide behaviours [18] but decreased the likelihood of self-harm [60], highlighting the need for a holistic approach when supporting military

communities. It is important to develop an enhanced understanding about motivations for engaging in self-harm and suicide behaviours, as for some self-harm acts as a coping mechanism to regulate internal emotions, whereas suicide behaviours become more apparent when the individual can no longer cope [73].

One key risk factor for self-harm was having left service. Potentially the transition from serving to ex-serving is a period of risk and support should be in place during this time. It might be that after leaving service, ex-serving personnel no longer feel like part of the military 'family', experience a shift in their sense of self and have difficulty connecting to civilian life [74]. Alternatively, it could reflect the influence of time as ex-serving personnel are typically older, and therefore, if considering lifetime prevalence and risk, this would capture a longer period of time. Additionally, there may be an underrepresentation among serving personnel as they are potentially more hesitant to report these behaviours due to stigma and fear of negative consequences to their career [75]; personnel with ill health (physical or mental) may be forced to leave service (i.e. medical discharge) and those who involuntarily separate from the military are at high risk of self-harm and suicide behaviours [22,76,77].

Poor health was another key identified risk factor, in particular clinical or probable diagnosis of PTSD, depression and anxiety. This highlights the importance of early detection of mental health problems, providing adequate care and support to military personnel throughout their military career, and providing continuity of care as they transition out of service. Interestingly, seeking help from formal medical sources was positively associated with lifetime suicide attempts [18] and taking more than five years to seek help was negatively associated with suicidal ideation [59]. It is unlikely that seeking help from formal medical sources has a causal relationship with suicide behaviours. Instead, UK research has found that ex-serving personnel are known to present to clinical services at times of crisis, which might involve an active episode of self-harm or suicide behaviours, therefore, placing them more at risk [78]. Additionally, those who delay seeking help for longer may have discovered ways to cope with their difficulties on their own, whereas those who seek help sooner may have been in crisis and at higher risk of suicidal ideation [59]. Clearly, a key challenge relating to self-harm and suicide behaviour risk is prevention. Most developed nations already have suicide prevention strategies in place, including for serving and ex-serving personnel, or as part of the wider mental health strategy [79–82]. The findings of this review suggest that prevention and intervention strategies should focus on the broader context of improving health and wellbeing (physical, mental, and social health).

Less research focussed on protective factors for self-harm and suicide behaviours among the included military populations. Despite this, one key association was with higher levels of perceived social support [18,44], indicating that this was likely a salient factor that could help prevent or mitigate risk, particularly due to its modifiable nature. This aligns with other international military populations, including the US [83], as well as in the general population where social support has been recommended for use in suicide prevention strategies [84]. One way to bolster social support among military populations is the use of peer support as a preventative strategy [85,86].

## 4.1. Strengths and limitations

A strength of this review was the search of seven literature databases using a broad search strategy outlined in an a priori PROSPERO approved review protocol. The protocol was generally followed but any changes were reflected by updating the protocol (i.e., removing the limit around the age of the sample). Additionally, a proportion of the eligibility assessment and critical appraisal of papers was conducted by a second, independent reviewer with high inter-rater

reliability. Including a wider range of geographical regions may have led to more included papers (e.g., Na et al (2021) [87]), however focus this review aimed to address the literature gap by collating evidence where attention has previously been limited.

There were some limitations to note. As with all systematic reviews, the findings of this review were subject to publication bias. Additionally, some potential associated factors received less attention than others, however this review tried to provide the best synthesis of the evidence currently available.

There were also several limitations relating to the included papers. First, papers used different definitions and measurements of self-harm and suicide behaviours, limiting the possibility of comparing findings across studies. However, combining these papers in a review contributes to understanding. Second, there is no universally agreed definition of veterans (i.e., ex-serving personnel) which makes cross-cultural comparison difficult. Third, studies did not always report on some military characteristics (such as service branch, rank, deployment experience, era of service and time since leaving service) and demographic characteristics (such as ethnicity and sexuality) which would have been useful for contextualisation and interpretation of the findings. There remains limited understanding of the impact of sex, experiences of ethnic minority and LGBTQ+ personnel. Fourth, included studies used a variety of sample sizes (range n = 144 to 233,803), study procedures, and sample characteristics. For example, the majority of papers relied on self-reported data which may have been subject to recall bias and social desirability bias. Finally, the included papers did not allow for meta-analyses to be conducted due to heterogeneity in the populations and outcomes.

## 4.2. Implications

This review has several important implications for policy, practice, and research. The identification of risk and protective factors can be useful to inform military health services and policies including where to target suicide prevention policy to reduce the incidence and impact of suicide. In the UK, this is one of nine key health priority themes laid out in the Defence People Health and Wellbeing Strategy 2022–2027 [88].

Identifying the risk and protective factors for self-harm and suicide behaviour outcomes is an important aspect of the development and implementation of effective prevention and intervention strategies to protect the mental health and wellbeing of military populations. Evidence on associated factors can inform healthcare practice and service provision for the Armed Forces community by highlighting several at-risk groups which may require additional support. This review suggests that additional support is required during the period of transition from military to civilian life but also highlights the importance of prevention early on in military service to reduce the impact on personnel as they transition out of service. The identified protective factors suggest that prevention and intervention strategies should focus on encouraging help-seeking for mental health problems early on before crisis events occur, as well as promoting social support networks and strengthening connections with family, friends and the community as a whole.

Additionally, focus should be placed on modifiable factors (i.e., factors that could reasonably be altered, such as psychiatric symptoms and social support). Although non-modifiable factors can help to identify level of risk, they are of less use during prevention and intervention as they cannot be changed to alter the level of risk. Therefore, particular focus on modifiable risk and protective factors is warranted as they are amenable to therapeutic intervention and can be key to addressing long-term risk due to their adaptive nature. However, it is still important to understand the role of non-modifiable factors as there may be an indirect effect, for example when considering military rank, it might be that officers are at lower risk of self-harm

and suicide behaviours because they typically hold higher socio-economic status and higher educational attainment [89] which are known protective factors.

Future studies should also focus on conducting longitudinal investigations which distinguish between pre-, peri- and post-service factors in order to identify pathways of self-harm and suicide behaviours, and to ensure support is in place at the right point in the military lifecycle.

## 5. Conclusions

This review highlighted several risk and protective factors for self-harm and suicide behaviours which warrant attention. Adequate care and support are a necessity for serving and ex-serving personnel who may be at risk of experiencing self-harm or suicide behaviours. Particular focus should be placed on implementing prevention strategies early on in military service to reduce the impact on personnel as they transition out of service. The identified protective factors suggest that prevention and intervention strategies should promote social networks as a key source of support for military personnel. Whilst this review was limited due to the paucity of peer-reviewed research within some populations, current work, such as that being undertaken in New Zealand will add to the understanding. Research should continue to progress towards understanding and preventing self-harm and suicide behaviours among military populations.

## Supporting information

**S1 File.** S1. Concepts and Keywords Used in Database Searches. S2. Key Findings of Included Papers. S3. List of Included Papers. S4. Quality Assessment. S5. PRISMA Checklist. (DOCX)

## Acknowledgments

We would like to thank Colonel Clare Bennett, Chief Mental Health Officer New Zealand Defence Force, for sharing relevant work being conducted in New Zealand.

## Author Contributions

**Conceptualization:** Charlotte Williamson.

**Formal analysis:** Charlotte Williamson, Bethany Croak.

**Funding acquisition:** Nicola T. Fear, Sharon A. M. Stevelink.

**Investigation:** Charlotte Williamson, Bethany Croak.

**Methodology:** Charlotte Williamson, Marie-Louise Sharp, Sharon A. M. Stevelink.

**Project administration:** Charlotte Williamson.

**Supervision:** Marie-Louise Sharp, Sharon A. M. Stevelink.

**Validation:** Charlotte Williamson, Bethany Croak.

**Visualization:** Charlotte Williamson.

**Writing – original draft:** Charlotte Williamson.

**Writing – review & editing:** Bethany Croak, Amos Simms, Nicola T. Fear, Marie-Louise Sharp, Sharon A. M. Stevelink.

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
