## [Decision Letter · Decision Letter 0]

15 Dec 2023

PONE-D-23-35992Risk and Protective Factors for Self-Harm and Suicide Behaviours among Serving and Ex-Serving Personnel of the UK Armed Forces, Canadian Armed Forces, Australian Defence Force and New Zealand Defence Force: A Systematic ReviewPLOS ONE

Dear Dr. Williamson,

Thank you for submitting your manuscript to PLOS ONE. After careful consideration, we feel that it has merit but does not fully meet PLOS ONE’s publication criteria as it currently stands. Therefore, we invite you to submit a revised version of the manuscript that addresses the points raised during the review process. Please submit your revised manuscript by Jan 29 2024 11:59PM. If you will need more time than this to complete your revisions, please reply to this message or contact the journal office at plosone@plos.org. Please include the following items when submitting your revised manuscript:A rebuttal letter that responds to each point raised by the academic editor and reviewer(s). You should upload this letter as a separate file labeled 'Response to Reviewers'.A marked-up copy of your manuscript that highlights changes made to the original version. You should upload this as a separate file labeled 'Revised Manuscript with Track Changes'.An unmarked version of your revised paper without tracked changes. You should upload this as a separate file labeled 'Manuscript'.If applicable, we recommend that you deposit your laboratory protocols in protocols.io to enhance the reproducibility of your results. Protocols.io assigns your protocol its own identifier (DOI) so that it can be cited independently in the future. For instructions see: https://journals.plos.org/plosone/s/submission-guidelines#loc-laboratory-protocols. Additionally, PLOS ONE offers an option for publishing peer-reviewed Lab Protocol articles, which describe protocols hosted on protocols.io. Read more information on sharing protocols at https://plos.org/protocols?utm_medium=editorial-email&utm_source=authorletters&utm_campaign=protocols.

We look forward to receiving your revised manuscript.

Kind regards,

Darrell Eugene Singer, M.D., M.P.H.

Academic Editor

PLOS ONE

Journal Requirements:

   "CW is currently in receipt of a funded PhD studentship via Phase 4 of the King’s Centre for Military Health Research Health and Wellbeing Cohort Study funded by the Office for Veterans’ Affairs (OVA), Cabinet Office, UK Government. BC is currently in receipt of a funded PhD studentship from The Economic and Social Research Council (ESRC), UK Government. AS is a full-time member of the British Army seconded to King’s College London. NTF is partly funded by the United Kingdom’s Ministry of Defence (MOD). MLS is fully funded by a grant from the OVA. SAMS is supported by the National Institute for Health and Care Research (NIHR) Maudsley Biomedical Research Centre at South London and Maudsley NHS Foundation Trust and the National Institute for Health and Care Research, NIHR Advanced Fellowship, Dr Sharon Stevelink, NIHR300592. The views expressed in this publication are those of the authors and not necessarily those of the OVA, the ESRC, the British Army, the MOD, the NHS, or the NIHR." 

Additional Editor Comments:I found your manuscript interesting and well-written.  In addition to responding to the reviewers' comments, please also address the following points:

Review the submission guidelines  at https://journals.plos.org/plosone/s/submission-guidelinesInclude page numbers and line numbers in the manuscript file. Use continuous line numbers (do not restart the numbering on each page).Please review the formatting guidelines (click here) and address errors that include, but are not limited to the following: section headings should not contain numbers; left justify all headings; remove boldface from figure and table references; and use fonts as directed in the formatting guidelines).Please review your references and reference guidelines. A review of the first ten references revealed one link “not found” (#7) and an incomplete reference (#10). Online references should include “Available from”, not “Available.” #7 should also include the date cited.

Reviewers' comments:

Reviewer's Responses to Questions

**Comments to the Author**

1. Is the manuscript technically sound, and do the data support the conclusions?

Reviewer #1: Yes

Reviewer #2: Yes

2. Has the statistical analysis been performed appropriately and rigorously? 

Reviewer #1: N/A

Reviewer #2: Yes

3. Have the authors made all data underlying the findings in their manuscript fully available?

Reviewer #1: Yes

Reviewer #2: No

4. Is the manuscript presented in an intelligible fashion and written in standard English?

Reviewer #1: Yes

Reviewer #2: Yes

5. Review Comments to the Author

Reviewer #1: This systematic review fills an important gap in describing risk and protective factors for self harm behaviors in those in non-United States 'Five Eyes' military populations and it is recommended to approve for publication with minimal changes.

Content Considerations:

1. The methods section does an excellent job of describing search strategy, study selection, and data extraction; however, more discussion of the synthesis methods used as applicable may provide additional insight into the measures to be presented in the results and discussion section.

Formatting/Grammar Considerations:

None identified

Reviewer #2: Please add a supplementary table with the results of the Quality Assessment Tool for Observational Cohort and Cross-Sectional Studies for the 28 included studies.

Abstract

Results, what is the difference between “early service leavers” and “shorter length of service?”

Conclusion, second sentence, I suggest revising “should” to “may.” As you know association is not causation.

Background

1st paragraph, last sentence, can you provide an estimate/reference for suicide behaviors specifically in serving versus ex-serving personnel, as opposed to a global prevalence in the entire (serving only?) military? Ex-serving personnel is not a common term at least in U.S. military literature. Is there a reason you do not use the term veteran?

2nd paragraph, please define or give examples of “social support.”

Page 5, can you provide a reference for the term “Five Eyes Alliance?”

Method

Was the search strategy peer-reviewed?

Did you consider searching the Cochrane library?

At least one expert in what, from each nation was contacted?

Did you consider cross-referencing for additional papers of relevance from published systematic reviews, which were excluded?

Data extraction on page 8, please provide the categories of the data extracted (e.g., age mean, median, rank categories, number of suicide attempts). What are “associated factors?”

Add a statement of what type of synthesis you plan to do and why you will not do meta-analyses.

Since papers were not excluded based on their quality, did you consider doing sensitivity analyses?

Results

I suggest you replace supplementary 2 with table 1.

Quality Assessment, please include a supplementary table of the results (yes, no, other) for the 14 criteria in the NIH QA tool for each of the 28 included papers. I expected to see criteria 14 (Were key potential confounding variables measured and adjusted statistically for their impact on the relationship between exposure(s) and outcome(s)?) to also be a common reason for lower quality scores.

Page 11, section 3.3.1, second paragraph, spell out ICD.

Page 12, section 3.4, I suggest you revise to read “Numerous risk and protective factors were identified in the review with some and some not statistically significant.” Section 3.4.1 to 3.5.2, where possible how the exposure variables (e.g., age, number of years after leaving service, education, rank) and the comparison variables were defined.

On page 13 instead of saying “…around two to seven times…” use the term “ranged from…to…” 2nd paragraph is the term ”junior rank” limited to enlisted or does it also include officers?

Page 14, 1st paragraph, does “lifetime suicidal ideation” refer to ex-service?

Section 3.5.1 does “middle/older age groups” apply to age at time of assessment for protective factors or for self-harm? How was “longer length of service” defined?

Page 15, revise to “…higher versus lower educational attainment and being employed versus unemployed.” I found it counter-intuitive that higher cumulative duration of Afghanistan-related deployment and taking longer than five years to seek support were protective factors. I think you offered adequate potential explanations for both in the discussion.

Discussion

I know you excluded U.S. studies and gave a rationale for this decision, but it would be informative if your review found risk and protective factors similar to the U.S. literature.

Page 19, add to your limitations that the included papers did not allow for meta-analyses due to heterogeneity in the populations and outcomes.

Table 2

Page 43, please verify that “Civilian in-religious terror” is a protective factor.

Page 54, please verify that “Suicide Ideation: Health: Panic Disorder” is a protective factor.

6. PLOS authors have the option to publish the peer review history of their article (what does this mean?). If published, this will include your full peer review and any attached files.

Reviewer #1: No

Reviewer #2: **Yes: **David W. Niebuhr

---

## [Author Response · Author response to Decision Letter 0]

27 Jan 2024

Dear Editor and Reviewers,

We would like to thank you for the useful comments on our manuscript and for the opportunity to revise and resubmit this paper to PLOS ONE. We have made changes to the manuscript based on reviewers’ comments and made some clarifications to reviewer queries below. We believe this improves the quality of the manuscript and makes it acceptable for publication in PLOS ONE. Please find our responses to each individual reviewer comment below. We have included page numbers where appropriate, and these page numbers refer to the clean copy of the manuscript. The changes within the manuscript have been marked using tracked changes for your convenience.

Reviewer #1:

1. This systematic review fills an important gap in describing risk and protective factors for self-harm behaviors in those in non-United States 'Five Eyes' military populations and it is recommended to approve for publication with minimal changes.

Thank you for taking the time to review the manuscript and for your recommendation to approve the manuscript for publication. We have addressed the suggested changes below.

2. The methods section does an excellent job of describing search strategy, study selection, and data extraction; however, more discussion of the synthesis methods used as applicable may provide additional insight into the measures to be presented in the results and discussion section. 

Thank you for your positive feedback around the methods section. To improve further, we have added a section to the methods covering data synthesis. Please see section 2.5 (pages 9-10) which reads - ‘To examine the risk and protective factors associated with self-harm and suicide behaviours, a narrative synthesis was performed. This approach was chosen due to heterogeneity across studies, for example, variation in sample sizes, target populations and outcomes [41]. Due to variation across included studies, such as differences in study design and outcome measurements, and the large number of risk and protective factors identified, it was neither practical nor feasible to conduct a meta-analysis as part of this review.’ 

Reviewer #2: 

1. Please add a supplementary table with the results of the Quality Assessment Tool for Observational Cohort and Cross-Sectional Studies for the 28 included studies.

We have added a table to present the full results of the quality assessment – please see Supplementary 4 (pages 95-96). 

Abstract

2. Results, what is the difference between “early service leavers” and “shorter length of service?

“Early service leavers (ESLs)” is a specific term usually defined as those who left service before completing the minimum term of service, in the UK this is 4 years. Shorter length of service varies from study to study depending on what information is presented, for instance, in one study (Bergman et al, 2019) the results are presented for untrained ESL, trained ESL, 4-6 years, 7-9 years, 10-12 years, 13-16 years, 17-22 years and ≥23 years’ service. In this instance, significant findings were found for both ESL groups, shorter length of service but not ESL (4-6 and 7-9 years), and longer length of service (≥23 years service). With ESL and shorter length of service being risk factors, and longer length of service being protective. We have tried to make this clearer in the results section of the manuscript. For example, ‘Other military characteristics that increased the risk of self-harm included being an early service leaver, defined within the relevant papers as <4 years [54] and ≤2.5 years [62] of service. This was also apparent for those with a shorter length of service (but not an early service leaver), for instance 4 to 6 years of service [62]’ (page 13). We have also rephrased the abstract to state ‘Identified risk factors included being single/ex-relationship, early service leavers, shorter length of service (but not necessarily early service leavers)…’ (page 2). 

3. Conclusion, second sentence, I suggest revising “should” to “may.” As you know association is not causation. 

As suggested by the reviewer we have amended this wording, please see page 3.

Background

4. 1st paragraph, last sentence, can you provide an estimate/reference for suicide behaviors specifically in serving versus ex-serving personnel, as opposed to a global prevalence in the entire (serving only?) military? 

We have added an additional sentence to the end of this paragraph to provide the prevalence for serving vs ex-serving personnel. This now reads - ‘Suicide behaviours can present as ideation (i.e. thoughts) and attempts which are also prevalent among serving and ex-serving personnel; a reported 11% global prevalence of these behaviours in the entire military [13], with prevalence varying slightly between serving and ex-serving personnel for both suicidal ideation (serving = 10%; ex-serving = 14%) and suicide attempts (serving = 8%; ex-serving = 15%) [13]’ (page 4).

5. Ex-serving personnel is not a common term at least in U.S. military literature. Is there a reason you do not use the term veteran?

We have chosen to use the term ex-serving personnel because there is no universally agreed definition of veterans (i.e. ex-serving personnel). The variation in terminology has been highlighted within the limitations section of the manuscript (pages 20) – ‘Second, there is no universally agreed definition of veterans (i.e., ex-serving personnel) which makes cross-cultural comparison difficult’. The term veteran can be used differently across nations, we have added a paragraph to pages 7-8 explaining this in more detail - ‘It is important to note the term ‘veteran’ is defined and used differently across nations. For instance, in the UK a veteran is defined as someone who has served a minimum of one day paid employment in the UK Armed Forces but no longer serves [38], this can include those who deployed or not and is irrespective of their type of discharge. However, in Canada, a veteran is any former member of the Canadian Armed Forces who successfully underwent basic training and is honourably discharged [39]. Therefore, for the purpose of this review, we adopted the term ‘ex-serving personnel’ to mean someone who served in the Armed Forces but no longer serves, irrespective of deployment experience or type of discharge’.

We chose not to adopt the US terminology because as this review explores four of the Five Eyes nations, excluding the US, therefore felt the US terminology may not be the most helpful or appropriate in this instance. 

6. 2nd paragraph, please define or give examples of “social support.”

Social support is defined within the psychological and epidemiological literature as the availability and adequacy of social connections, comprised of structural (received) and functional (perceived) support. Structural social support refers to the presence of relationships, which includes objective measures such as size of social network, whilst functional social support incorporates quality of support, for example how successfully a relationship fulfils ones needs. Sources of support can include friends, family, partners, and colleagues. 

To make this clearer in the manuscript, we have amended text to read ‘… but include social support (i.e. the availability and adequacy of social connections [25])’ (pages 4-5).

7. Page 5, can you provide a reference for the term “Five Eyes Alliance?” 

As requested, a reference has been added. Please see reference number 30. 

Method

8. Was the search strategy peer-reviewed?

The full search strategy was reviewed by all team members prior to conducting the search.

9. Did you consider searching the Cochrane library?

The Cochrane library was not searched as part of this review, but relevant papers should have been identified through the databases searched and cross-referencing the reference lists of included papers.

10. At least one expert in what, from each nation was contacted?

This refers to an expert in military self-harm/suicide behaviour research. For instance, the expert in New Zealand was the Chief Mental Health Officer New Zealand Defence Force. All of the experts contacted are currently involved in the Five Eyes Mental Health Research and Innovation Collaboration (Five Eyes MHRIC - https://annual-report.cimvhr.ca/impact/five-eyes). We have reworded the mauscript to make this clearer (pages 6-7) – ‘At least one expert in the field of military self-harm and/or suicide behaviour research from each nation was contacted to ensure key papers had been identified. Experts were identified via the Five Eyes Mental Health Research and Innovation Collaboration [37].’

11. Did you consider cross-referencing for additional papers of relevance from published systematic reviews, which were excluded?

Cross-referencing was performed using the reference lists of studies included in the review and other relevant studies, including the reference lists of any identified systematic-reviews and meta-analyses of relevance. We have added clarification in the text on page 6 which now reads ‘Additionally, the reference lists of included studies and other relevant studies were searched, including identified systematic reviews and meta-analyses of relevance)’. 

12. Data extraction on page 8, please provide the categories of the data extracted (e.g., age mean, median, rank categories, number of suicide attempts). What are “associated factors?”

As suggested by the reviewer, we have added some further details regarding the data extraction (see page 9). For example: ‘Sample characteristics: age in years (mean, median or range), sex distribution, ethnicity, population type (e.g., clinical/non-clinical). Military characteristics: serving status (serving/ex-serving), engagement type (regular/reserve), service branch (Army, Naval Services, Air Force), rank (other, non-commissioned, commissioned), deployment experience (e.g., number and duration of deployments), era of service (e.g., Iraq and Afghanistan era)’. 

We have also reworded the term ‘associated factors’ to provide clarity. This now reads ‘Risk and protective factors associated with self-harm and/or suicide behaviour outcomes’. Please see page 9.

13. Add a statement of what type of synthesis you plan to do and why you will not do meta-analyses.

We have added a section to the methods covering data synthesis. Please see section 2.5 (pages 9-10). Please see Reviewer 1, Comment 2 for more information.

14. Since papers were not excluded based on their quality, did you consider doing sensitivity analyses?

Thank you for raising this. Indeed, we did consider performing a sensitivity analysis. However, our findings indicated that around three-quarters of papers were being rated as ‘fair’ quality and one-quarter as ‘good’ quality. This means that if we were to do a deeper dive to make a comparison between the findings of good versus fair quality papers, it would be difficult to produce meaningful findings as not all papers explored the same risk and protective factors or the same self-harm/suicide behaviour outcomes. 

Results

15. I suggest you replace supplementary 2 with table 1.

We have decided not to swap these tables as Supplementary 2 is far too dense (28 pages long) to be included in the main body of the paper, and would take away from the findings presented in Table 2. 

16. Quality Assessment, please include a supplementary table of the results (yes, no, other) for the 14 criteria in the NIH QA tool for each of the 28 included papers. I expected to see criteria 14 (Were key potential confounding variables measured and adjusted statistically for their impact on the relationship between exposure(s) and outcome(s)?) to also be a common reason for lower quality scores.

As per Reviewer 2, Comment 1 We have added a table to present the full results of the quality assessment – please see Supplementary 4 (pages 95-96). The expectation surrounding criteria 14 is true for a couple of papers but the majority did adjust for key confounders. 

17. Page 11, section 3.3.1, second paragraph, spell out ICD.

We have now spelled out International Classification of Diseases. Please see page 12, section 3.3.1.

18. Page 12, section 3.4, I suggest you revise to read “Numerous risk and protective factors were identified in the review with some and some not statistically significant.” 

We have amended this sentence to read ‘Numerous risk and protective factors were identified in the review with varying levels of statistical significance [Table 2]’. Please see page 13, section 3.4. 

19. Section 3.4.1 to 3.5.2, where possible how the exposure variables (e.g., age, number of years after leaving service, education, rank) and the comparison variables were defined.

Where possible, more detail has been added to sections 3.4.1 to 3.5.2 to define exposure/comparison variables. Please see pages 13-16. For example, ‘Higher levels of perceived social support [18,44] (e.g., a score of 33-36 on the multidimensional scale of perceived social support [18])…’. 

20. On page 13 instead of saying “…around two to seven times…” use the term “ranged from…to…”. 2nd paragraph is the term ”junior rank” limited to enlisted or does it also include officers?

We have amended the wording to read ‘ranging from two to seven times more likely’. Please see page 14, section 3.4.2. Additionally, in this instance “more junior rank during service” varies slightly across studies. We have added an example for clarity (page 14) – ‘holding a more junior rank during service [43,44,61] (e.g., non-commissioned officer or other lower ranks [43,44])’.

21. Page 14, 1st paragraph, does “lifetime suicidal ideation” refer to ex-service?

Lifetime suicidal ideation refers to an experience of suicidal at any point in their lifetime (including before/during/after service). In the Jones et al (2019) study, this was found among serving and ex-serving personnel. We have tried to make this clearer in the text, ‘Further, lifetime suicidal ideation was reported as a risk factor for lifetime suicide attempts among a sample of serving and ex-serving personnel (up to 12 times more likely) [18]’ (page 15). 

22. Section 3.5.1 does “middle/older age groups” apply to age at time of assessment for protective factors or for self-harm? How was “longer length of service” defined?

That is correct, in this instance age group refers to the age of participant at assessment (e.g., age at questionnaire completion). Additionally, “longer length of service” varies slightly across studies. We have added an example for clarity (page 15) – ‘…and having a longer length of service [15,62] (e.g., ≥23 years [62])’. 

23. Page 15, revise to “…higher versus lower educational attainment and being employed versus unemployed.” 

The wording has been revised to read ‘…higher educational attainment (vs lower educational attainment) [48,49] (e.g., higher than Year 10 [48,49]), and being employed (vs unemployed) [29].’ Please see section 3.5.2, page 16.

24. I found it counter-intuitive that higher cumulative duration of Afghanistan-related deployment and taking longer than five years to seek support were protective factors. I think you offered adequate potential explanations for both in the discussion.

We agree with the reviewer that some of the findings seem counterintuitive but appreciate that you think the potential explanations provided in the discussion are adequate. 

Discussion

25. I know you excluded U.S. studies and gave a rationale for this decision, but it would be informative if your review found risk and protective factors similar to the U.S. literature.

We have added a line to the discussion to highlight the similarities in risk factors identified in this systematic review and US literature. Please see page 17 – ‘Several of the identified risk factors reflect those among the general population [3,70], serving and ex-serving military personnel of the US Armed Forces [27,33], and other similar occupational groups (such as emergency responders) [71], and were generally consistent across the included geographical regions. Some similar identified risk factors for suicide among the US military community include mental health diagnoses (e.g. mood, alcohol/substance, psychotic and personality disorders), and experience of childhood adversi

---

## [Editor Report · Decision Letter 1]

8 Feb 2024

Risk and Protective Factors for Self-Harm and Suicide Behaviours among Serving and Ex-Serving Personnel of the UK Armed Forces, Canadian Armed Forces, Australian Defence Force and New Zealand Defence Force: A Systematic Review

PONE-D-23-35992R1

Dear Dr. Williamson,

We’re pleased to inform you that your manuscript has been judged scientifically suitable for publication and will be formally accepted for publication once it meets all outstanding technical requirements.

Kind regards,

Darrell Eugene Singer, M.D., M.P.H.

Academic Editor

PLOS ONE

Additional Editor Comments:  Dr. Williamson: my thanks and congratulations to you and your co-authors on addressing the reviewers concerns and your collective patience with our U.S. military perspective. Your polite and direct comments are appreciated.   

There are a few formatting edits that need to be made, but the remainder of the editorial process will catch those (over another minor revision).  Best on your manuscript and future research!  Regards, Darrell Singer